# Adversarially Robust Spiking Neural Networks Through Conversion

**Ozan Özdenizci**                                                      *ozan.ozdenizci@igi.tugraz.at*
*Institute of Theoretical Computer Science, Graz University of Technology, Graz, Austria*
*TU Graz - SAL Dependable Embedded Systems Lab, Silicon Austria Labs, Graz, Austria*

**Robert Legenstein**                                                *robert.legenstein@igi.tugraz.at*
*Institute of Theoretical Computer Science, Graz University of Technology, Graz, Austria*

**Reviewed on OpenReview:** *https://openreview.net/forum?id=I8FMYa2BdP*

## Abstract

Spiking neural networks (SNNs) provide an energy-efficient alternative to a variety of artificial neural network (ANN) based AI applications. As the progress in neuromorphic computing with SNNs expands their use in applications, the problem of adversarial robustness of SNNs becomes more pronounced. To the contrary of the widely explored end-to-end adversarial training based solutions, we address the limited progress in scalable robust SNN training methods by proposing an adversarially robust ANN-to-SNN conversion algorithm. Our method provides an efficient approach to embrace various computationally demanding robust learning objectives that have been proposed for ANNs. During a post-conversion robust finetuning phase, our method adversarially optimizes both layer-wise firing thresholds and synaptic connectivity weights of the SNN to maintain transferred robustness gains from the pre-trained ANN. We perform experimental evaluations in a novel setting proposed to rigorously assess the robustness of SNNs, where numerous adaptive adversarial attacks that account for the spike-based operation dynamics are considered. Results show that our approach yields a scalable state-of-the-art solution for adversarially robust deep SNNs with low-latency.

## 1 Introduction

Spiking neural networks (SNNs) are powerful models of computation based on principles of biological neuronal networks (Maass, 1997). Unlike traditional artificial neural networks (ANNs), the neurons in an SNN communicate using binary signals (i.e., spikes) elicited across a temporal dimension. This characteristic alleviates the need for computationally demanding matrix multiplication operations of ANNs, thus resulting in an advantage of energy-efficiency with event-driven processing capabilities (Roy et al., 2019). Together with the developments in specialized neuromorphic hardware to process information in real-time with low-power requirements, SNNs offer a promising energy-efficient technology to be widely deployed in safety-critical real world AI applications (Davies et al., 2021). There are two main streams of applications for SNNs. First, they are applied directly to spiking input streams, e.g., from dynamic vision sensors (Bellec et al., 2018; Wu et al., 2019; Kim & Panda, 2021a). Second, they are applied to standard machine learning tasks and standard machine learning data sets (Rathi & Roy, 2021; Bu et al., 2022; Jiang et al., 2023). We focus in this article on the latter case. Here, conversion methods that convert a pre-trained ANN to an SNN have been shown to yield SNN models with excellent performance (Diehl et al., 2015; Roy et al., 2019). Post-conversion finetuning can then be used for low-latency and energy-efficient SNNs (Rathi & Roy, 2021; Davies et al., 2021).

Numerous studies have explored the susceptibility of traditional ANNs to adversarial attacks (Szegedy et al., 2013), and this security concern evidently extends to SNNs since effective attacks are gradient-based

and architecture agnostic (Sharmin et al., 2019). Recently, there has been growing interest in achieving adversarial robustness with SNNs, which remains an important open problem (Kundu et al., 2021; Ding et al., 2022; Liang et al., 2022). Temporal operation characteristics of SNNs makes it non-trivial to apply existing defenses developed for ANNs (Madry et al., 2018). Nevertheless, current solutions are mainly end-to-end adversarial training based methods that are adapted for SNNs (Ding et al., 2022). These approaches, however, yield limited robustness for SNNs as opposed to ANNs, both due to their computational limitations (e.g., infeasibility of heavily iterative backpropagation across time), as well as the confounding impact of spike-based end-to-end adversarial training, on the SNN gradients, as we elaborate later in our evaluations. According to the current literature, the widely accepted assumption is that SNNs directly trained with spike-based backpropagation achieve stronger robustness than SNNs obtained by converting a pre-trained ANN (Sharmin et al., 2020). In this work, we challenge this hypothesis by proposing a novel adversarially robust ANN-to-SNN conversion algorithm based solution.

We present an ANN-to-SNN conversion algorithm that improves SNN robustness by exploiting adversarially trained baseline ANN weights at initialization followed by an adversarial finetuning method. Our approach allows to integrate any existing robust learning objective developed for conventional ANNs into an ANN-to-SNN conversion algorithm, thus optimally transferring robustness gains into the SNN. Specifically, we convert a robust ANN into an SNN by adversarially finetuning both the synaptic connectivity weights and the layer-wise firing thresholds, which yields significant robustness benefits. Innovatively, we introduce a highly effective approach to incorporate adversarially pre-trained ANN batch-norm layer parameters – which helps facilitating adversarial training of the deep ANN – within spatio-temporal SNN batch-norm operations, without the need to omit these layers from the models as conventionally done in conversion based methods.

Previous works on improving adversarial robustness of SNNs conduct attack evaluations in a similar way to ANNs (Sharmin et al., 2020; Kundu et al., 2021; Ding et al., 2022), which are insufficient due to not considering the influence of the spike function surrogate gradient used by the adversary. To that end, we also introduce a novel ensemble SNN robustness evaluation approach where we rigorously simulate the attacks by constructing adaptive adversaries based on different differentiable approximation techniques (see Section 4), accounting for the architectural dynamics of SNNs that are different from ANNs. By doing so, we adapt recent observations on evaluating ANNs with non-differentiable operations and/or obfuscated gradients (Athalye et al., 2018a; Carlini et al., 2019) to the domain of SNNs, and improve SNN robustness evaluations with rigor.

In depth experimental analyses show that our approach achieves a scalable state-of-the-art solution for adversarial robustness in deep SNNs with low-latency, outperforming recently introduced end-to-end adversarial training based algorithms with up to $2\times$ larger robustness gains and reduced robustness-accuracy trade-offs.

## 2 Preliminaries and Related Work

### 2.1 Spiking Neural Networks

Unlike traditional ANNs with continuous valued activations, SNNs use discrete, pulsed signals to transmit information. Discrete time dynamics of leaky-integrate-and-fire (LIF) neurons in feed-forward SNNs follow:

$$\mathbf{v}^l(t^-) = \tau\, \mathbf{v}^l(t-1) + \mathbf{W}^l\, \mathbf{s}^{l-1}(t), \tag{1}$$

$$\mathbf{s}^l(t) = H(\mathbf{v}^l(t^-) - V_{th}^l), \tag{2}$$

$$\mathbf{v}^l(t) = \mathbf{v}^l(t^-)(1 - \mathbf{s}^l(t)), \tag{3}$$

where the neurons in the $l$-th layer receive the output spikes $\mathbf{s}^{l-1}(t)$ from the preceding layer weighted by the trainable synaptic connectivity matrix $\mathbf{W}^l$, and $\tau$ denotes the membrane potential leak factor. We represent the neuron membrane potential before and after the firing of a spike at time $t$ by $\mathbf{v}^l(t^-)$ and $\mathbf{v}^l(t)$, respectively. $H(.)$ denotes the Heaviside step function. A neuron $j$ in layer $l$ generates a spike when its membrane potential $v_j^l(t^-)$ reaches the firing threshold $V_{th}^l$. Subsequently, the membrane potential is updated via Eq. (3), which performs a *hard-reset*. Alternatively, a neuron with *soft-reset* updates its membrane potential by subtraction, $\mathbf{v}^l(t) = \mathbf{v}^l(t^-) - \mathbf{s}^l(t)V_{th}^l$, where the residual membrane potential can encode information. Inputs $\mathbf{s}^0(t)$ to the first layer are typically represented either by *rate coding*, where the input signal intensity determines the

firing rate of an input Poisson spike train of length $T$, or by *direct coding*, where the input signal is applied as direct current to $\mathbf{v}^1(t^-)$ for all $T$ timesteps of the simulation.

**Direct Training:** One category of algorithms to obtain deep SNNs relies on end-to-end training with spike-based backpropagation (Wu et al., 2018; 2019; Lee et al., 2020). Different from conventional ANNs, gradient-based SNN training backpropagates the error through the network temporally (i.e., backpropagation through time (BPTT)), in order to compute a gradient for each forward pass operation. During BPTT, the discontinuous derivative of $H(\mathbf{z}^l(t))$ for a spiking neuron when $\mathbf{z}^l(t) = \mathbf{v}^l(t^-) - V_{th}^l \geq 0$, is approximated via surrogate gradient functions (Neftci et al., 2019). Such approximate gradient based learning algorithms have reduced the performance gap between deep ANNs and SNNs over the past years (Bellec et al., 2018; Shrestha & Orchard, 2018). Various improvements proposed for ANNs, e.g., residual connections or batch normalization, have also been adapted to SNNs (Fang et al., 2021; Duan et al., 2022). Notably, recent work introduced batch-norm through time (BNTT) (Kim & Panda, 2021b), and threshold-dependent batch-norm (tdBN) (Zheng et al., 2021) mechanisms to better stabilize training of deep SNNs.

**ANN-to-SNN Conversion:** An alternative approach is to convert a structurally equivalent pre-trained ANN into an SNN (Cao et al., 2015). This is performed by replacing ANN activation functions with LIF neurons (thus keeping weights $\mathbf{W}^l$ intact), and then calibrating layer-wise firing thresholds (Diehl et al., 2015; Sengupta et al., 2019). The main goal is to approximately map the continuous-valued ANN neuron activations to the firing rates of spiking neurons (Rueckauer et al., 2017). While conversion methods leverage prior knowledge from the source ANN, they often need more simulation time steps $T$ than BPTT-based SNNs which can temporally encode information. Recent, *hybrid* approaches tackle this problem via post-conversion finetuning (Rathi et al., 2020). This was later improved with soft-reset mechanisms (Han et al., 2020), or trainable firing thresholds and membrane leak factors during finetuning (Rathi & Roy, 2021). More recent methods redefine the source ANN activations to minimize conversion loss (Li et al., 2021; Bu et al., 2022).

## 2.2 Adversarial Attacks and Robustness

Deep neural networks are known to be susceptible to adversarial attacks (Szegedy et al., 2013), and various defenses have been proposed to date. Currently, most effective ANN defense methods primarily rely on exploiting adversarial examples during optimization (Athalye et al., 2018a), which is known as adversarial training (AT) (Madry et al., 2018; Goodfellow et al., 2015).

Given samples $(\mathbf{x}, y)$ from a training dataset $\mathcal{D}$, a neural network $f_{\boldsymbol{\theta}}$ is conventionally trained via the natural cross-entropy loss $\mathcal{L}_{\text{CE}}(f_{\boldsymbol{\theta}}(\mathbf{x}), y)$, which follows a maximum likelihood learning rule. In the context of robust optimization, adversarial training objectives can be formulated by:

$$\min_{\boldsymbol{\theta}} \ \mathbb{E}_{(\mathbf{x},y)\sim\mathcal{D}} \left[ \max_{\tilde{\mathbf{x}} \in \Delta_\epsilon^p(\mathbf{x})} \mathcal{L}_{\text{robust}} \left( f_{\boldsymbol{\theta}}(\tilde{\mathbf{x}}), f_{\boldsymbol{\theta}}(\mathbf{x}), y \right) \right], \tag{4}$$

with $\tilde{\mathbf{x}}$ denoting the adversarial example crafted during the inner maximization step, which is constrained within $\Delta_\epsilon^p(\mathbf{x}) := \{\tilde{\mathbf{x}} : \|\tilde{\mathbf{x}} - \mathbf{x}\|_p \leq \epsilon\}$, defined as the $l_p$-norm ball around $\mathbf{x}$ with an $\epsilon > 0$ perturbation budget. We will focus on $l_\infty$-norm bounded examples where $\tilde{\mathbf{x}} \in \Delta_\epsilon^\infty(\mathbf{x})$.

Standard AT (Madry et al., 2018) only utilizes iteratively crafted adversarial examples during optimization, i.e., $\mathcal{L}_{\text{robust}}^{\text{AT}} := \mathcal{L}_{\text{CE}}(f_{\boldsymbol{\theta}}(\tilde{\mathbf{x}}), y)$, thus maximizing log-likelihood solely based on $\tilde{\mathbf{x}}$, and does not use benign samples. Conventionally, the inner maximization is approximated by projected gradient descent (PGD) on a choice of an $\mathcal{L}_{\text{PGD}}$ loss (e.g., for standard AT: $\mathcal{L}_{\text{PGD}} = \mathcal{L}_{\text{CE}}(f_{\boldsymbol{\theta}}(\tilde{\mathbf{x}}), y)$), to obtain $k$-step adversarial examples by: $\tilde{\mathbf{x}}_{k+1} = \Pi_{\Delta_\epsilon^\infty(\mathbf{x})} \left[ \tilde{\mathbf{x}}_k + \eta \cdot \text{sign}(\nabla_{\tilde{\mathbf{x}}_k} \mathcal{L}_{\text{PGD}}(f_{\boldsymbol{\theta}}(\tilde{\mathbf{x}}_k), y)) \right]$, where $\tilde{\mathbf{x}}_0 = \mathbf{x} + \delta$ with $\delta \sim \mathcal{U}(-\epsilon, \epsilon)$, step size $\eta$, and $\Pi_{\Delta_\epsilon^\infty(\mathbf{x})}[.]$ the clipping function onto the $\epsilon$-ball. Due to its iterative computational load, single-step variants of PGD, such as the fast gradient sign method (FGSM) (Goodfellow et al., 2015) or random-step FGSM (RFGSM) (Tramèr et al., 2018), have been explored as alternatives (Wong et al., 2020).

State-of-the-art robust training methods propose regularization schemes to improve the inherent robustness-accuracy trade-off (Tsipras et al., 2019). A powerful robust training loss, TRADES (Zhang et al., 2019a), redefines the training objective as:

$$\mathcal{L}_{\text{robust}}^{\text{TRADES}} := \mathcal{L}_{\text{CE}}(f_{\boldsymbol{\theta}}(\mathbf{x}), y) + \lambda_{\text{TRADES}} \cdot D_{\text{KL}} \left( f_{\boldsymbol{\theta}}(\tilde{\mathbf{x}}) \| f_{\boldsymbol{\theta}}(\mathbf{x}) \right), \tag{5}$$

where $D_{\mathrm{KL}}$ denotes the Kullback-Leibler divergence, and $\lambda_{\mathrm{TRADES}}$ is the trade-off hyper-parameter. In a subsequent work (Wang et al., 2020), MART robust training loss proposes to emphasize the error on misclassified examples via:

$$\mathcal{L}_{\mathrm{robust}}^{\mathrm{MART}} := \mathcal{L}_{\mathrm{BCE}}(f_{\boldsymbol{\theta}}(\tilde{\mathbf{x}}), y) \ + \lambda_{\mathrm{MART}} \cdot (1 - f_{\boldsymbol{\theta}}^{(y)}(\mathbf{x})) \cdot D_{\mathrm{KL}}\left(f_{\boldsymbol{\theta}}(\tilde{\mathbf{x}}) || f_{\boldsymbol{\theta}}(\mathbf{x})\right), \tag{6}$$

where $\lambda_{\mathrm{MART}}$ is the trade-off parameter, $f_{\boldsymbol{\theta}}^{(y)}(\mathbf{x})$ denotes the probability assigned to class $y$, and the boosted cross-entropy is defined as, $\mathcal{L}_{\mathrm{BCE}}(f_{\boldsymbol{\theta}}(\tilde{\mathbf{x}}), y) := \mathcal{L}_{\mathrm{CE}}(f_{\boldsymbol{\theta}}(\tilde{\mathbf{x}}), y) - \log(1 - \max_{j \neq y} f_{\boldsymbol{\theta}}^{(j)}(\tilde{\mathbf{x}}))$, with the second term improving the decision margin for adversarial samples.

### 2.3 Adversarial Robustness in Spiking Neural Networks

Earlier explorations of adversarial attacks on SNNs particularly emphasized that direct training exhibits stronger robustness compared to conversion-based SNNs with high latency, and rate input coding SNNs tend to yield higher robustness (Sharmin et al., 2019; 2020). Accordingly, subsequent work mainly focused on end-to-end SNN training methods, yet relying on shallower networks and simpler datasets (Sharmin et al., 2020; Marchisio et al., 2020). Notably, SNNs were also shown to maintain higher resiliency to black-box attacks than their ANN counterparts (Sharmin et al., 2020). Later studies analyzed various components of SNNs and their robustness implications, such as the structural configuration (El-Allami et al., 2021; Kim et al., 2022a), or the surrogate gradient used for training (Xu et al., 2022). Another line of work focuses on designing attacks for event-based neuromorphic data, with stable BPTT gradient estimates (Marchisio et al., 2021; Liang et al., 2021; Büchel et al., 2022). Until recently, there has been limited progress in scalable robust SNN training algorithms.

A recent work (Liang et al., 2022) explores certified adversarial robustness bounds for SNNs on MNIST-variant datasets, based on an interpretation of methods designed for ANNs (Zhang et al., 2019b). Recently, HIRE-SNN (Kundu et al., 2021) proposed adversarial finetuning of SNNs initialized by converting a naturally trained ANN, using a similar approach to (Rathi & Roy, 2021). For finetuning, HIRE-SNN manipulates inputs temporally with single-step adversarial noise, resulting in marginal robustness benefits on relatively small models. The recently proposed SNN-RAT (Ding et al., 2022) algorithm empirically achieved state-of-the-art robustness with SNNs. SNN-RAT interprets the ANN weight orthogonalization methodology to constrain the Lipschitz constant of a feed-forward network (Cisse et al., 2017), in the context of SNNs optimized with single-step adversarial training. Due to its ability to scale to deep SNN architectures, SNN-RAT currently serves as a solid baseline to achieve robustness with SNNs.

## 3 Adversarially Robust ANN-to-SNN Conversion

### 3.1 Converting Adversarially Trained Baseline ANNs

Our method converts an adversarially trained ANN into an SNN by initially configuring layer-wise firing thresholds based on the well-known threshold-balancing approach (Sengupta et al., 2019). Then we propose to adversarially finetune these thresholds together with the weights.

**Configuration of the SNN:** Given an input $\mathbf{x}$, we use a direct input encoding scheme for $\mathbf{s}^0(t)$, where the input signal (e.g., pixel intensity) is applied as a constant current for a simulation length of $T$ timesteps. Subsequently, the network follows the SNN dynamics from Eqs. (1), (2), (3). We mainly consider SNNs with integrate-and-fire (IF) neurons, i.e., $\tau = 1$, similar to Ding et al. (2022) (see *Experiments on TinyImageNet* for LIF neuron simulations). The output layer of our feed-forward SNN with $L$ layers only accumulates weighted incoming inputs without generating a spike. Integrated non-leaky output neuron membrane potentials $v_j^L(T)$ for $j = 1, \ldots, N$, with $N$ being the number of classes, are then used to estimate normalized probabilities via:

$$f_{\boldsymbol{\theta}}^{(j)}(\mathbf{x}) = e^{v_j^L(T)} \Big/ \sum_{i=1}^{N} e^{v_i^L(T)}, \quad \text{where} \quad \mathbf{v}^L(t) = \mathbf{v}^L(t-1) + \mathbf{W}^L \mathbf{s}^{L-1}(t), \ \ t = 1, \ldots, T, \tag{7}$$

and $f_{\boldsymbol{\theta}}^{(j)}(\mathbf{x})$ denotes the probability assigned to class $j$. In our case, the neural network parameters $\boldsymbol{\theta}$ to be optimized will include connectivity weights, layer-wise firing thresholds, and batch-norm parameters.

**Initializing SNN Weights:** We use the weights from a baseline ANN that is adversarially trained via Eq. (4), to initialize the SNN synaptic connectivity matrices. Our approach is in principle agnostic to the ANN adversarial training method. For a feed-forward ANN, neuron output activations $\mathbf{a}^l$ in the $l$-th layer would be, $\mathbf{a}^l = \sigma(\mathbf{W}^l \mathbf{a}^{l-1})$, where $\sigma(.)$ often indicates a ReLU. We essentially maintain $\mathbf{W}^l$ for all layers, and replace $\sigma(.)$ with spiking neuron dynamics.

**Conversion of Batch Normalization:** Traditional conversion methods either exclude batch-norm from the source ANN (Sengupta et al., 2019), or absorb its parameters into the preceding convolutional layer (Rueckauer et al., 2017). Here, we introduce a novel and highly effective approach to directly integrate robustly pre-trained ANN batch normalization affine transformation parameters into SNN threshold-dependent batch-norm (tdBN) layers (Zheng et al., 2021), and propose to maintain the parameters $\varphi^l$ and $\omega^l$ within:

$$\hat{\mathbf{S}}^l = \varphi^l \cdot \left( (\bar{\mathbf{S}}^l - \mathbb{E}[\bar{\mathbf{S}}^l]) \Big/ \sqrt{Var[\bar{\mathbf{S}}^l] + \vartheta} \right) + \omega^l, \quad \text{where} \quad \bar{\mathbf{S}}^l = [\bar{\mathbf{s}}^l(1), \bar{\mathbf{s}}^l(2), \dots, \bar{\mathbf{s}}^l(T)], \tag{8}$$

weighted spiking inputs $\bar{\mathbf{s}}^l(t) = \mathbf{W}^l \mathbf{s}^{l-1}(t)$ are gathered temporally in $\bar{\mathbf{S}}^l$, $\vartheta$ is a tiny constant, and $\mathbb{E}[.]$ and $Var[.]$ are estimated both across time and mini-batch. This temporal and spatial normalization was found highly effective in stabilizing training. Normalized $\hat{\mathbf{s}}^l(t)$ is then integrated into the neuron membrane potential via Eq. (1). In order to accurately maintain the utility of the preserved $\varphi^l$ and $\omega^l$ parameters for the SNN, we do not re-use the moving average mean and variance estimates from the ANN, which are often tracked to be used at inference time. Instead, we discard these previous statistics, and re-estimate them during SNN finetuning. Different from its original formulation (Zheng et al., 2021), we do not perform a threshold-based scaling in this normalization, but instead finetune $\varphi^l$ and $\omega^l$ together with the firing thresholds.

**Initialization of Trainable Firing Thresholds:** We generate inputs $\mathbf{s}^0(t)$ by direct coding for $T_c$ calibration time steps, which we choose to be much larger than $T$, using a number of training set mini-batches. In order to estimate proportionally balanced per-layer firing thresholds as in (Sengupta et al., 2019), starting from the first layer, for all inputs we record the pre-activation values (i.e., summation of the weighted spiking inputs) observed across $T_c$ timesteps, and set the threshold to be the maximum value in the $\rho$-percentile of the distribution of these pre-activation values (often $\rho > 99\%$). This is performed similarly to (Lu & Sengupta, 2020; Rathi & Roy, 2021), in order to accommodate for the stochasticity involved due to the use of sampled training set mini-batches (see Suppl. A.2 for more details).

After setting the firing threshold for $l = 1$, we perform a forward pass through the spiking neuron dynamics of this layer, and move on to the next layer with spiking activations. We sequentially set the firing thresholds for all other layers in the same way. During the forward passes, we only use inference-time statistics estimated spatio-temporally on the current batch for normalization via tdBN. After estimating all layer thresholds, we initialize the trainable $\{V_{th}^l\}_{l=1}^{L-1}$ values by scaling the initial estimates with a constant factor $\kappa < 1$, in order to promote a larger number of spikes to flow across layers proportionally to our initial estimates (Lu & Sengupta, 2020) (see Suppl. A.3 for complete algorithm).

## 3.2 Robust Finetuning of SNN After Conversion

We complement our hybrid conversion approach with spike-based backpropagation based adversarial finetuning. We exploit adversarial examples for robustly pre-trained weight finetuning updates, and also to robustly adjust firing thresholds. Our finetuning objective utilizes the TRADES loss (Zhang et al., 2019a):

$$\min_{\boldsymbol{\theta}} \mathbb{E} \left[ \mathcal{L}_{\text{CE}}(f_{\boldsymbol{\theta}}(\mathbf{x}), y) + \beta \cdot \max_{\tilde{\mathbf{x}} \in \Delta_\epsilon^p(\mathbf{x})} D_{\text{KL}}(f_{\boldsymbol{\theta}}(\tilde{\mathbf{x}}) || f_{\boldsymbol{\theta}}(\mathbf{x})) \right], \tag{9}$$

where $\beta$ is a robustness-accuracy trade-off parameter, $\boldsymbol{\theta}$ consists of $\{\mathbf{W}^l, \varphi^l, \omega^l\}_{l=1}^L$ and $\{V_{th}^l\}_{l=1}^{L-1}$, and $f_{\boldsymbol{\theta}}(\tilde{\mathbf{x}})$ and $f_{\boldsymbol{\theta}}(\mathbf{x})$ are the probability vectors assigned to a benign and an adversarial example, estimated by the integrated output neuron membrane potentials over $T$ timesteps. In order to minimize this objective, the regularizer term requires an adversarial example $\tilde{\mathbf{x}}$ obtained through an inner maximization step. This adversarial example $\tilde{\mathbf{x}} \in \Delta_\epsilon^\infty(\mathbf{x})$ is obtained via an RFGSM based single inner maximization step by:

$$\tilde{\mathbf{x}} = \mathbf{x}' + (\epsilon - \alpha) \cdot \text{sign}(\nabla_{\mathbf{x}'} \mathcal{L}_{\text{RFGSM}}), \quad \text{where} \quad \mathbf{x}' = \mathbf{x} + \alpha \cdot \text{sign}(\mathcal{N}(\mathbf{0}, \mathbf{I})). \tag{10}$$

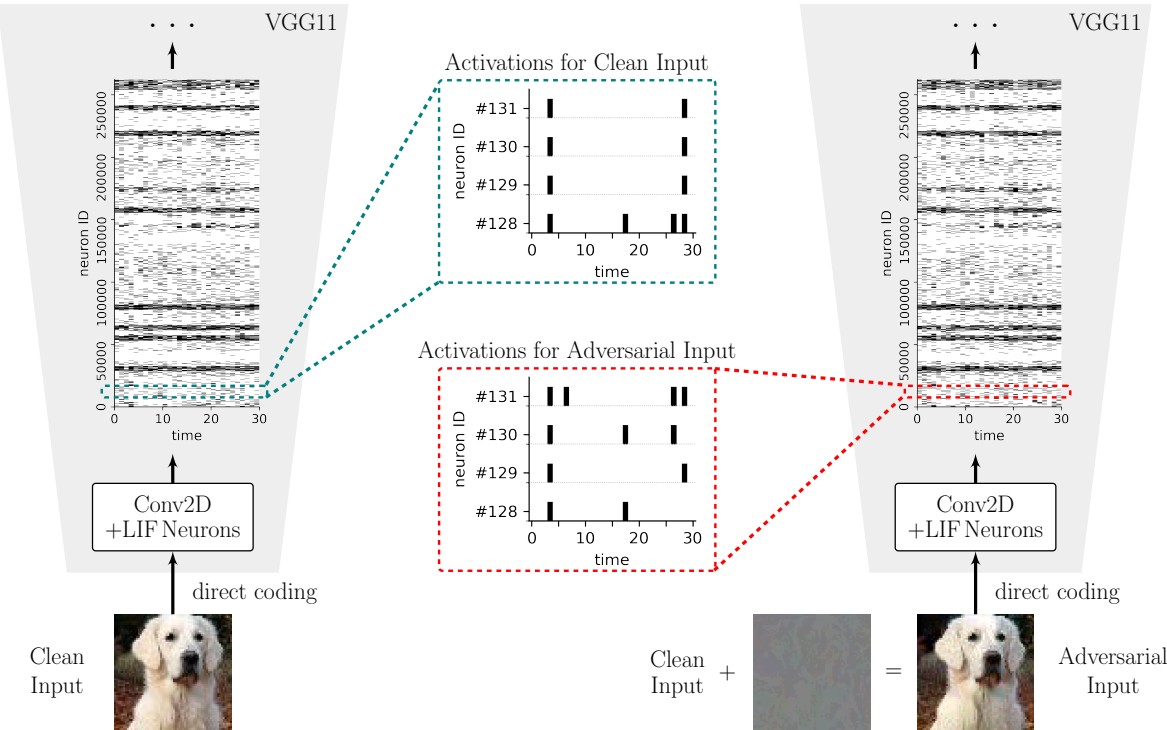

Figure 1: Illustration of the impact of an adversarial attack on feed-forward SNNs with direct input coding. In this setting, the spiking representation of the input is generated after the first convolutional layer, where the input pixel intensities are applied as direct current to the neurons for $T$ time steps. We show one clean/adversarial test case from TinyImageNet against our VGG11 model ($T = 30$) obtained via conversion. Left side depicts the clean input scenario, and the right side depicts an adversarial attack scenario. The first layer of this model consists of 64 Conv2D kernels, which outputs 262,144 LIF neuron activations (64 feature maps with 64×64 spatial image resolution). Adversarial perturbations of magnitude $\epsilon = 8/255$ (negative values are visualized darker and positive values are visualized brighter) are obtained in the input image pixel domain via spike-based BPTT. The change in the input is hardly visible for the adversarial image. However, this small change leads to a noticeable difference of the spiking output of the first convolutional layer on the fine temporal scale (see cyan and red zoom-in boxes). Neuron IDs are vectorized for illustration purposes.

Note that only a single gradient computation is used for RFGSM, thus it is a computationally efficient and yet a powerful alternative to PGD (Ding et al., 2022). For RFGSM, differently from its original formulation with a cross-entropy loss, we propose to use single-step gradient ascent for $\mathcal{L}_{\mathrm{RFGSM}} = D_{\mathrm{KL}}(f_{\boldsymbol{\theta}}(\mathbf{x}')||f_{\boldsymbol{\theta}}(\mathbf{x}))$, which maximizes the KL-divergence between the vectors of integrated membrane potentials of output class neurons (see Suppl B.3 ablations for alternatives). Objective in Eq. (9) is optimized via BPTT, and a piecewise linear surrogate gradient with unit dampening factor is used (Bellec et al., 2018). We present an outline of our BPTT-based parameter updates, as well as our overall training algorithm with further details in Suppl. A.3.

## 4   An Ensemble Evaluation Approach for Adversarial Robustness of SNNs

We focus on low-latency SNNs via conversion, which are widely applicable to standard machine learning tasks. Therefore we mainly use SNNs with direct input coding, where the input pixel intensities are applied as direct current to the first layer LIF neurons, and a spiking representation of the input is generated through their activations. In this SNN setting, adversarial attacks operate similar as in the ANN setting, where adversarial perturbations are obtained by computing input gradients through spike-based backpropagation through time, and then applied in the input pixel domain. These adversarial examples then result in a different spiking activation pattern after the first convolutional layer of the network, as illustrated in Fig. 1.

Spike-based backpropagation relies on surrogate gradient functions due to the discontinuous derivative of the spike activation function (Neftci et al., 2019). Thus, differently from ANNs, the success of a backpropagation based SNN adversarial attack also critically depends on the surrogate gradient used by the adversary for the attack. Previous studies on adversarial robustness of SNNs (Sharmin et al., 2020; Kundu et al., 2021; Ding et al., 2022) conduct naive attack evaluations, by only considering the same surrogate gradient function used during training at the time of attack. Here, we propose to adapt recent observations made on ANNs with non-differentiable operations and/or obfuscated gradients (Brendel & Bethge, 2017; Athalye et al., 2018a; Carlini et al., 2019) to the domain of SNNs, in order to improve SNN robustness evaluations by potentially relying on more stable gradient estimates with different surrogate gradient choices at the time of attack.

For evaluations, we therefore implemented an ensemble attack strategy. For each attack algorithm, we anticipate an adaptive SNN adversary, as proposed for ANNs (Carlini et al., 2019), that simulates a number of attacks for a test sample where the BPTT surrogate gradient varies in each case. We consider variants of the three most common surrogate gradient functions: piecewise linear (Bellec et al., 2018), exponential (Shrestha & Orchard, 2018), and rectangular (Wu et al., 2018), with different width, dampening, or steepness parameters (see Suppl. A.4 for parameter details). In the ensemble we also consider straight-through estimation (Bengio et al., 2013), backward pass through rate, and a conversion-based approximation (Ding et al., 2022), where we replace spiking neurons with a ReLU during BPTT. We calculate robust accuracies on the whole test set, and consider an attack successful for a test sample if the model is fooled with any of the attacks from the ensemble. Note that during training we only use the piecewise linear surrogate gradient with a unit dampening factor as in Ding et al. (2022).

## 5 Experiments

### 5.1 Datasets and Models

We performed experiments with CIFAR-10, CIFAR-100, SVHN and TinyImageNet datasets. We used VGG11 (Simonyan & Zisserman, 2015), ResNet-20 (He et al., 2016) and WideResNet (Zagoruyko & Komodakis, 2016) architectures with depth 16 and width 4 (i.e., WRN-16-4) in our experiments (see Suppl. A.1 for further details). Unless specified otherwise, SNNs were run for $T = 8$ timesteps as in Ding et al. (2022).

### 5.2 Robust Training Configurations

**Baseline ANNs:** We performed adversarially robust ANN-to-SNN conversion with baseline ANNs optimized via state-of-the-art robust training objectives: standard AT (Madry et al., 2018), TRADES (Zhang et al., 2019a), and MART (Wang et al., 2020), as well as ANNs optimized with natural training. We mainly focus on adversarial training with a perturbation bound of $\epsilon_1 = 2/255$. In addition, for CIFAR-10 we also performed conversion of ANNs with standard AT under $\epsilon_2 = 4/255$ and $\epsilon_3 = 8/255$. This resulted in baseline ANNs with: AT-$\epsilon_1$, TRADES-$\epsilon_1$, MART-$\epsilon_1$, AT-$\epsilon_2$, AT-$\epsilon_3$.

**Converted SNNs:** Robust finetuning was performed for 60 epochs (only for CIFAR-100 we used 80 epochs). Regardless of the baseline ANN training scheme, SNN finetuning was always performed with $\epsilon_1$ bounded perturbations. At test time, we evaluated robustness also at higher perturbation levels $\epsilon_2$ and $\epsilon_3$, similar to Ding et al. (2022). We used $\beta = 2$ for CIFAR-10, and $\beta = 4$ in all other experiments. Further configurations are detailed in Suppl. A.2. Our implementations can be found at: https://github.com/IGITUGraz/RobustSNNConversion.

### 5.3 Adversarial Robustness Evaluations

**White-box Adversarial Attacks:** In accordance with the ensemble setting presented in Section 4, we evaluate SNNs against FGSM (Goodfellow et al., 2015), RFGSM (Tramèr et al., 2018), PGD (Madry et al., 2018), and Auto-PGD attacks based on cross-entropy loss (APGD$_{CE}$) and difference of logits ratio loss (APGD$_{DLR}$) (Croce & Hein, 2020). We also consider a targeted version of this attack (T-APGD$_{CE}$), where each sample was adversarially perturbed towards 9 target classes from CIFAR-10 or CIFAR-100, until successful. PGD attack step size $\eta$ is determined via $2.5 \times \epsilon/\#$steps (Madry et al., 2018). For baseline ANNs, we present our evaluations via AutoAttack (Croce et al., 2020) at $\epsilon_3$ (see Table A1 for extended evaluations).

Table 1: Robustness evaluations (%) of baseline ANNs and our converted SNNs across different datasets and architectures. Robust accuracies of converted SNNs are evaluated separately under $\epsilon_1 = 2/255$, $\epsilon_2 = 4/255$, $\epsilon_3 = 8/255$ perturbation budgets, with white-box ensemble SNN attacks. Robust accuracies of baseline ANNs are presented here with AutoAttack for an $\epsilon_3$ perturbation budget (see Table A1 for other ANN attacks)

| Dataset & Architecture | ANN Training Objective | Baseline ANN Clean / $\epsilon_3$-Robust Acc. | Adversarially Robust ANN-to-SNN Conversion | | | | | | |
|---|---|---|---|---|---|---|---|---|---|
| | | | Clean Acc. | $\epsilon_1$-Robust Acc. | | $\epsilon_2$-Robust Acc. | | $\epsilon_3$-Robust Acc. | |
| | | | | FGSM | $PGD^{20}$ | FGSM | $PGD^{20}$ | FGSM | $PGD^{20}$ |
| CIFAR-10 with VGG11 | Natural | 95.10 / 0.00 | 93.53 | 66.62 | 59.22 | 42.81 | 22.44 | 15.71 | 0.74 |
| | **AT-$\epsilon_1$** | 93.55 / 16.64 | 92.14 | 74.10 | 70.02 | 59.11 | 46.59 | 33.94 | 13.18 |
| | **TRADES-$\epsilon_1$** | 92.10 / 27.62 | 91.39 | 75.01 | 72.00 | 62.64 | 51.47 | 41.42 | 19.65 |
| | **MART-$\epsilon_1$** | 92.68 / 21.98 | 91.47 | 74.83 | 70.35 | 62.87 | 49.51 | 44.39 | 17.79 |
| CIFAR-100 with WRN-16-4 | Natural | 78.93 / 0.00 | 70.87 | 41.16 | 35.49 | 26.08 | 13.87 | 11.25 | 1.40 |
| | **AT-$\epsilon_1$** | 74.90 / 7.05 | 70.02 | 47.91 | 44.73 | 34.74 | 27.41 | 18.60 | 7.64 |
| | **TRADES-$\epsilon_1$** | 73.87 / 8.43 | 67.84 | 45.48 | 42.73 | 33.62 | 26.78 | 18.13 | 7.79 |
| | **MART-$\epsilon_1$** | 72.61 / 11.04 | 67.10 | 45.99 | 43.46 | 34.03 | 27.40 | 18.93 | 8.90 |
| SVHN with ResNet-20 | Natural | 97.08 / 0.08 | 94.18 | 77.38 | 74.30 | 59.93 | 48.09 | 31.50 | 10.35 |
| | **AT-$\epsilon_1$** | 96.50 / 23.58 | 91.90 | 74.15 | 71.80 | 60.66 | 52.68 | 36.60 | 21.51 |
| | **TRADES-$\epsilon_1$** | 96.29 / 27.16 | 92.78 | 76.38 | 74.27 | 62.45 | 55.59 | 37.76 | 23.96 |
| | **MART-$\epsilon_1$** | 96.16 / 27.65 | 92.71 | 76.76 | 74.82 | 62.60 | 55.52 | 37.90 | 24.11 |

**Black-box Adversarial Attacks:** We performed query-based Square Attack (Andriushchenko et al., 2020) evaluations with various numbers of limited queries (presented in Section 6.4). We also evaluated transfer-based attacks (Papernot et al., 2017) to verify that such black-box attacks succeed less often than our white-box evaluations (presented in Suppl. B.1).

# 6 Experimental Results

## 6.1 Evaluating Robustness of Adversarially Trained ANN-to-SNN Conversion

We present our results in Table 1 using FGSM and $PGD^{20}$ ensemble attacks – which are significantly stronger than the attacks used during robust finetuning – for models obtained by converting baseline ANNs optimized with natural and different adversarial training objectives. These results show that our approach is compatible with any baseline ANN robust training algorithm, and an initialization from a relatively more robust ANN proportionally transfers its robustness to the converted SNN. Although we use an identical robust finetuning procedure, converted SNNs that used weaker (natural training) ANNs show minimal robustness under ensemble attacks (e.g., $\epsilon_3$-$PGD^{20}$ for CIFAR-10, Natural: 0.74, AT-$\epsilon_1$: 13.18, TRADES-$\epsilon_1$: 19.65, MART-$\epsilon_1$: 17.79, similar results can be observed with CIFAR-100 and SVHN). Here, conversion of a natural ANN and finetuning is analogous to the HIRE-SNN approach by Kundu et al. (2021), where the finetuning objective and optimized parameters are slightly different, nevertheless the final robustness remains significantly weak.

Our finetuning objective and robustly adjusting $V_{th}^l$ also appear critical to maintain robustness. We elaborate this later in our ablation studies in Suppl. B.3. In brief, we observed that natural $\mathcal{L}_{CE}$ finetuning (Rathi & Roy, 2021), or using fixed $V_{th}^l$ during robust finetuning, yields worse robustness (e.g., from Table B5, $\epsilon_3$-$PGD^{20}$ when converting using natural $\mathcal{L}_{CE}$: 10.03, using Eq. (9) but with fixed $V_{th}^l$: 11.51, Ours (trainable $V_{th}^l$): 13.18).

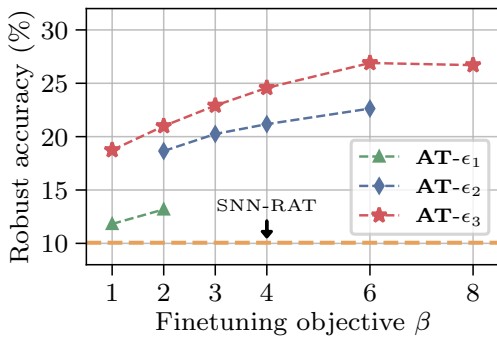

Figure 2: $\epsilon_3$-Robust accuracies under $PGD^{20}$ vs. $\beta$, with different baseline ANN standard AT perturbation levels.

Table 2: Comparisons with the state-of-the-art robust SNN training method SNN-RAT, based on the model checkpoints from Ding et al. (2022). All attacks are $l_\infty$-norm bounded with $\epsilon_3 = 8/255$, and evaluated through an ensemble. Note that FGSM, RFGSM and PGD[7] (with $\eta = 0.01$) are the only attacks evaluated in Ding et al. (2022), which we also include in ensemble setting.

| | | CIFAR-10 with VGG11 | | | CIFAR-100 with WRN-16-4 | | | |
| | | SNN-RAT | Conversion (Ours) | | | SNN-RAT | Conversion (Ours) | | |
| | | | AT-$\epsilon_1$ | TRADES-$\epsilon_1$ | MART-$\epsilon_1$ | | AT-$\epsilon_1$ | TRADES-$\epsilon_1$ | MART-$\epsilon_1$ |
|---|---|---|---|---|---|---|---|---|---|
| | Clean Acc. | 90.74 | **92.14** | 91.39 | 91.47 | 69.32 | **70.02** | 67.84 | 67.10 |
| $\epsilon_3$-Robust Acc. | FGSM | 32.80 | 33.94 | 41.42 | **44.39** | **19.79** | 18.60 | 18.13 | 18.93 |
| | RFGSM | 55.29 | 57.52 | 61.42 | **61.59** | 32.39 | **32.55** | 31.77 | 32.00 |
| | PGD[7] | 13.47 | 15.98 | **23.63** | 23.04 | 7.76 | 9.53 | 9.29 | **10.70** |
| | PGD[20] | 10.06 | 13.18 | **19.65** | 17.79 | 6.04 | 7.64 | 7.79 | **8.90** |
| | PGD[40] | 9.45 | 12.67 | **19.20** | 16.99 | 5.84 | 7.43 | 7.67 | **8.48** |
| | APGD$_{DLR}$ | 11.71 | 14.07 | **23.28** | 20.12 | 8.81 | 8.88 | 8.92 | **9.68** |
| | APGD$_{CE}$ | 9.07 | 10.05 | **17.63** | 15.54 | 5.22 | 6.62 | 6.94 | **7.67** |
| | T-APGD$_{CE}$ | 7.69 | 9.19 | **16.11** | 13.09 | 4.72 | 5.41 | 5.50 | **6.43** |

Fig. 2 shows the impact of baseline ANN adversarial training $\epsilon$, on the converted SNN, as well as the robustness-accuracy trade-off parameter $\beta$, for CIFAR-10. Note that we still perform SNN finetuning with $\epsilon_1$ as in Ding et al. (2022), since larger perturbations via BPTT yields an unstable finetuning process. We observe that baseline ANN robustness transfers also in this case, and one can obtain even higher SNN robustness (e.g., with AT-$\epsilon_3$, $\beta = 6$: 26.90%) by using our method via heavier adversarial training of the ANN where optimization stability is not a problem.

## 6.2 Comparisons to State-of-the-Art Robust SNNs with Direct Adversarial Training

In Table 2, we compare our approach with the current state-of-the-art method SNN-RAT, using the models and specifications from Ding et al. (2022), under extensive white-box ensemble attacks[1]. Our models yield both higher clean accuracy and adversarial robustness across various attacks, with some models being almost twice as robust against strong adversaries, e.g., clean/PGD[40] for CIFAR-10, SNN-RAT: 90.74/9.45, Ours (TRADES-$\epsilon_1$): 91.39/19.20. Note that in Table 2 we only compare our models trained through baseline ANNs with $\epsilon_1$-perturbations, since SNN-RAT also only uses $\epsilon_1$-adversaries (see Fig. 2 as to where SNN-RAT stands with respect to our models which can also exploit larger perturbation budgets via conversion). Overall, we observe that SNNs gradually get weaker against stronger adversaries, however our models maintain relatively higher robustness, also against powerful APGD$_{CE}$ and T-APGD$_{CE}$ attacks.

We further compare our approach with other direct AT methods (e.g., TRADES), within our ablation studies in Suppl. B.3, which again revealed worse performance than our conversion-based approach (e.g., from Table B5, clean/PGD[20] for CIFAR-10, direct TRADES training: 91.75/6.70, Ours (AT-$\epsilon_1$): 92.14/13.18).

**Experiments on TinyImageNet:** Results are summarized in Fig. 3. Here, we utilize the same VGG11 architecture used for BNTT (Kim & Panda, 2021b), which has claimed robustness benefits through its batch-norm mechanism combined with Poisson input encoding (see Suppl. A.4 for attack details on this model). For comparisons, besides our usual IF ($T = 8$) setting, we also converted (AT-$\epsilon_1$) ANNs by using LIF neurons ($\tau = 0.99$) and soft-reset with direct coding.

Results in Fig. 3b show that our model achieves a scalable, state-of-the-art solution for adversarially robust SNNs. Our model remains superior to RFGSM and SNN-RAT based end-to-end AT methods in the usual IF ($T = 8$) setting with 17.82% robust accuracy. Results with LIF neurons show that our models can improve clean accuracy and robustness with higher $T$ ($T = 30$)[2], whereas RFGSM and SNN-RAT perform generally

---

[1]The original work in Ding et al. (2022) reports a BPTT-based PGD[7] with 21.16% on CIFAR-10, and 11.31% on CIFAR-100 $\epsilon_3$-robustness. While we could confirm these results in their naive BPTT setting, this particular attack yielded lower SNN-RAT robust accuracies with our ensemble attack: 13.47% on CIFAR-10, and 7.76% on CIFAR-100.

[2]End-to-end adversarial SNN training, e.g., with SNN-RAT, for $T = 30$ was computationally infeasible using BPTT.

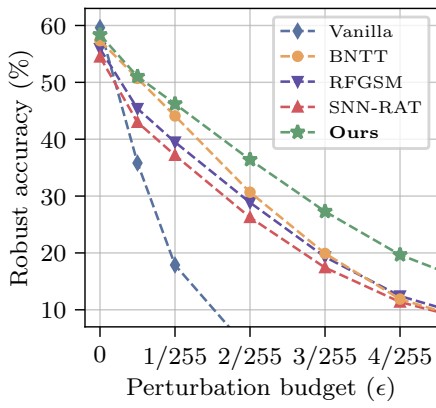

(a) Robustness under PGD[20] attacks.

| | Input Enc. | SNN Config. | Adv. Train | Clean Acc. | $\epsilon_2$-Robust Acc. | |
|---|---|---|---|---|---|---|
| | | | | | FGSM | PGD[20] |
| Vanilla | Direct | IF ($T=8$) | ✗ | 57.29 | 1.93 | 0.13 |
| RFGSM | Direct | IF ($T=8$) | ✓ | 54.29 | 19.21 | 14.55 |
| SNN-RAT | Direct | IF ($T=8$) | ✓ | 52.76 | 22.39 | 16.63 |
| **Ours (AT-$\epsilon_1$)** | Direct | IF ($T=8$) | ✓ | 55.98 | **22.91** | **17.82** |
| Vanilla | Direct | LIF ($T=30$) | ✗ | 59.62 | 1.76 | 0.05 |
| BNTT | Poisson | LIF ($T=30$) | ✗ | 57.33 | 17.64 | 11.88 |
| RFGSM | Direct | LIF ($T=8$) | ✓ | 55.92 | 17.51 | 12.42 |
| SNN-RAT | Direct | LIF ($T=8$) | ✓ | 54.47 | 19.66 | 11.37 |
| **Ours (AT-$\epsilon_1$)** | Direct | LIF ($T=8$) | ✓ | 57.21 | 22.52 | 16.25 |
| **Ours (AT-$\epsilon_1$)** | Direct | LIF ($T=30$) | ✓ | 58.36 | **24.02** | **19.67** |

(b) Comparisons with $\epsilon_2 = 4/255$ attacks.

Figure 3: Evaluations on TinyImageNet, with comparisons to SNN-RAT (Ding et al., 2022), an RFGSM based mixed AT baseline from Ding et al. (2022), BNTT (Kim & Panda, 2021b), and a vanilla SNN with natural training. SNNs with LIF neurons ($\tau = 0.99$) use soft-reset as in Kim & Panda (2021b). In (a) we compare LIF-neuron models from the bottom half of the table in (b), using Ours with $T = 30$.

worse with LIF neurons. Results in Fig. 3a also reveal stronger robustness of our LIF-based ($T = 30$) model starting from smaller perturbations, where it consistently outperforms existing methods. Our evaluations of BNTT show that AT is particularly useful for robustness (Ours ($T = 30$): 19.67%, BNTT: 11.88%). Although direct coding does not approximate inputs and attacks become simpler (Kim et al., 2022a), we show that direct coding SNNs can still yield higher resilience when adversarially calibrated.

## 6.3 Coding Efficiency of Adversarially Robust SNNs

In Fig. 4, we demonstrate higher coding efficiency of our approach. Specifically, VGG11 models consisted of 286,720 IF neurons operating for $T = 8$ timesteps, which could induce a total of approx. 2.3M possible spikes for inference on a single sample. For CIFAR-10, on average for a test sample (i.e., mean of histograms in Fig. 4a), the total #spikes elicited across a VGG11 was 148,490 with Ours, 169,566 for SNN-RAT, and 173,887 for a vanilla SNN. This indicates an average of only 6.47% active spiking neurons with our method, whereas SNN-RAT: 7.39%, and vanilla SNN: 7.58%. For CIFAR-100 in Fig. 4b, total #spikes elicited on average was 589,774 with Ours, 616,439 for SNN-RAT, and 592,297 for vanilla SNN, across approx. 3.8M possible spikes from the 475,136 neurons in WRN-16-4. Here, our model was 15.52% actively spiking, whereas 16.22% with SNN-RAT. We did not observe an efficiency change with our more robust SNNs, and spike activity was similar to the models converted from a baseline ANN with AT-$\epsilon_1$.

## 6.4 Further Experiments

**Query-based Black-box Robustness:** We evaluated SNNs with adversaries who have no knowledge of the defended model parameters, but can send a limited number of queries to it. Results are shown in Fig. 5, where our models consistently yield higher robustness (CIFAR-10 at 5000 queries: Ours: 50.2% vs. SNN-RAT: 46.3%, CIFAR-100: 18.8% vs. 17.5%). We also see that after 1000 queries, the baseline ANN shows weaker resilience than our SNNs. This reveals improved black-box robustness upon conversion to a spiking architecture, which is consistent with previous findings (Sharmin et al., 2020).

**Transfer-based Black-box Robustness:** Previous work (Athalye et al., 2018a) suggests that transfer-based attacks should be considered as a baseline to verify no potential influence of gradient masking or obfuscation. Accordingly, we present our evaluations in Suppl. B.1, where we verify (in Table B1) that fully black-box transfer attacks succeed less often than our ensemble white-box evaluations.

**Performance of Ensemble Attacks:** We elaborate the effectiveness of our ensemble attack approach in Suppl. B.2. In brief, we reveal that changing the shape or dampening magnitude of the surrogate gradient

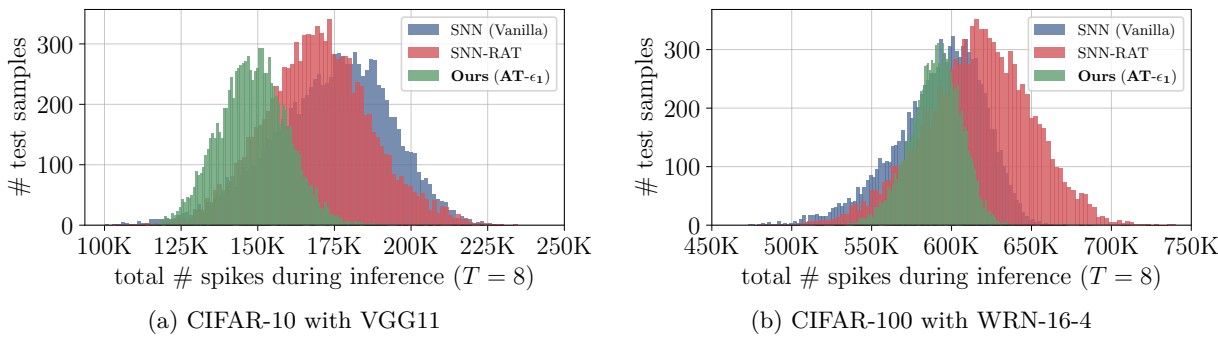

Figure 4: Coding efficiency comparisons of vanilla and adversarially robust SNNs.

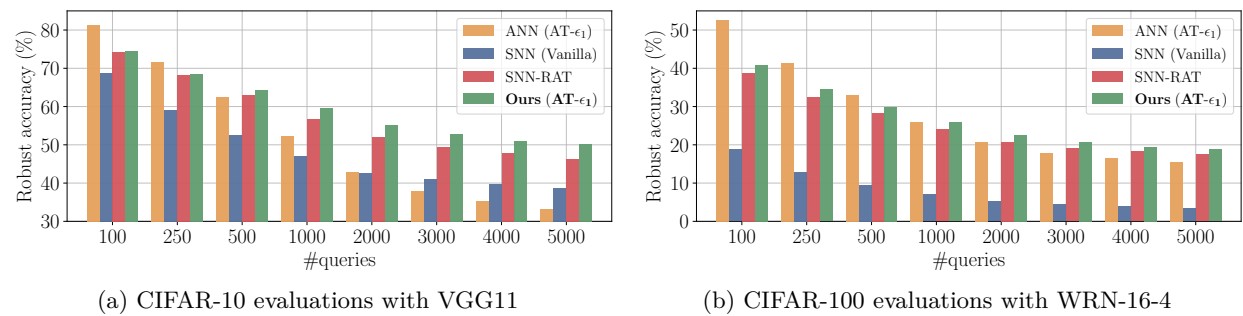

Figure 5: Square Attack (Andriushchenko et al., 2020) evaluations with limited number of queries for the baseline ANN (AT-$\epsilon_1$), end-to-end trained vanilla SNN, adversarially trained SNN-RAT, and ours (AT-$\epsilon_1$).

during BPTT can result in much stronger adversaries. Specifically, these attacks revealed stronger and more accurate robustness evaluations for SNNs directly optimized with AT (Ding et al., 2022). Moreover, we present a comparison of robust baseline ANNs and converted SNNs under analogous attacks in Table B3, where we highlight the need for an ensemble setting for SNNs, since naive evaluations can be misleading in relation to ANNs. We argue that our attacks reveal a more reliable robustness quantification for SNNs.

**Out-of-distribution Generalization:** We evaluated robust SNNs on CIFAR-10-C and CIFAR-100-C benchmarks (Hendrycks & Dietterich, 2019) in Suppl. B.4. Our analyses yielded mean CIFAR-10-C accuracies of 83.85% with our model, whereas SNN-RAT: 81.72%, Vanilla SNN: 81.22% (CIFAR-100-C, Ours: 54.06%, SNN-RAT: 51.95%, Vanilla SNN: 46.14%). We also observed that this result was consistent for all corruptions beyond noise (Patel et al., 2019) (see Tables B6 and B7).

## 7 Discussion

We propose an adversarially robust ANN-to-SNN conversion algorithm that exploits robustly trained baseline ANN weights at initialization, to improve SNN resiliency. Our method shows state-of-the-art robustness in feed-forward SNNs with low latency, outperforming recent end-to-end AT based methods in robustness, generalization, and coding efficiency. Our approach embraces any robust ANN training method to be incorporated in the domain of SNNs, and can also adopt further improvements to robustness, such as leveraging additional data (Rebuffi et al., 2021). Importantly, we empirically show that SNN evaluations against non-adaptive adversaries can be misleading (Carlini et al., 2019; Pintor et al., 2022), which has been the common practice in the field of SNNs.

We argue that direct adversarial training methods developed for ANNs yield limited SNN robustness both because of their computational limitations with BPTT (e.g., use of single-step inner maximization), and also our findings with BPTT-based adversarial training leading to SNNs with bias to the used surrogate gradient

and vulnerability against adversaries that can leverage this. Therefore we propose to harness stronger and stable adversarial training possibility through ANNs, and exploit better robustness gains.

One other innovation in this work is our method of utilizing adversarially pre-trained ANN batch-norm parameters within spiking tdBN layers, without the need to omit these from the models as conventionally done. Although batch-norm layers were found to help facilitating stable adversarial training of the deep ANNs that we consider for conversion, there has also been recent studies on ANNs that particularly explore the necessity of batch normalization layers in adversarial training. In particular, Benz et al. (2021) highlight the increasing adversarial vulnerability of ANNs that utilize batch normalization layers due to the models' emerging dependence on non-robust features. Similarly, Wang et al. (2022) proposes to improve robustness by removing batch-norm layers altogether during adversarial training, which is in principle restricted to deep residual network architectures that use stable weight initialization methods for well-behaved gradients during adversarial training. There are also more recent studies that propose to replace batch-norm layers in ANNs with more robust alternatives (e.g., aggregating several normalization layers and treating the model as an ensemble of different models (Dong et al., 2022)). Although there has been some findings regarding its positive contribution to robustness (Kim & Panda, 2021b), the role of batch normalization in SNN adversarial robustness is currently not fully explored and to be studied in future work. We use baseline ANNs that contained batch-norm layers in our simulations, since these architectures have been the commonly used ones (e.g., VGG architectures with batch-norm), especially in state-of-the-art ANN-to-SNN conversion studies (Rathi & Roy, 2021). In principle, our robust conversion algorithm still remains compatible with any of such developments even if the baseline ANN does not contain any batch-norm layers.

There were no observed white-box robustness benefits of SNNs with respect to ANNs. However, under black-box scenarios, our converted SNNs were more resilient. Overall, we achieved to significantly improve robustness of deep SNNs, considering their progress in safety-critical settings (Davies et al., 2021). Since SNNs offer a promising energy-efficient technology, we argue that their robustness should be an important concern to be studied for reliable AI applications.

### 7.1 Limitations

There are two main application scenarios for SNNs: (1) on event-based data, (2) standard non-event-based data sets where one can still achieve significant energy savings via SNNs (Davies et al., 2021). In our study we only focus on the latter, that is on direct coding SNNs with low latency where conventional attacks remain powerful as an important open problem. In this case, we can simply use a conventional ANN that is robustly pre-trained on the same task, which our method requires. On the other hand, DVS adversaries constitute a different problem setting tailored to the event-based domain where a pre-trained ANN is not available. This problem also requires consideration of specialized attacks where adversarial noise should be binary, hence the conventional $l_\infty$-norm attacks are not directly applicable (Marchisio et al., 2021; Büchel et al., 2022).

**Acknowledgments**

This work has been supported by the Graz Center for Machine Learning (GraML), and the "University SAL Labs" initiative of Silicon Austria Labs (SAL) and its Austrian partner universities for applied fundamental research for electronic based systems.

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

# A    Details on the Experimental Setup

## A.1    Datasets and Models

We experimented with CIFAR-10, CIFAR-100, SVHN and TinyImageNet datasets. CIFAR-10 and CIFAR-100 datasets both consist of 50,000 training and 10,000 test images of resolution 32×32, from 10 and 100 classes respectively (Krizhevsky, 2009). SVHN dataset consists of 73,257 training and 26,032 test samples of resolution 32×32 from 10 classes (Netzer et al., 2011). TinyImageNet dataset consists of 100,000 training and 10,000 test images of resolution 64×64 from 200 classes (Le & Yang, 2015). We use VGG11 (Simonyan & Zisserman, 2015), ResNet-20 (He et al., 2016) and WideResNet (Zagoruyko & Komodakis, 2016) models with depth 16 and width 4 (i.e., WRN-16-4).

For baseline ANN optimization, we adopted the conventional data augmentation approaches that were found beneficial in robust adversarial training (Rebuffi et al., 2021; Tack et al., 2022). During SNN finetuning after conversion, we follow the simple traditional data augmentation scheme that was also adopted in previous work (Ding et al., 2022). This procedure involves randomly cropping images into 32×32 dimensions by padding zeros for at most 4 pixels around it, and randomly performing a horizontal flip. For TinyImageNet we instead perform a random resized crop to 64×64, also followed by a random horizontal flip. All image pixel values are normalized using the conventional metrics estimated on the training set of each dataset.

## A.2    Robust Training Configurations

**Baseline ANNs:** During robust optimization with standard AT, TRADES, and MART, adversarial examples are iteratively crafted for each mini-batch (inner maximization step of Eq. (4)), by using 10 PGD steps with random starts under $l_\infty$-bounded perturbations, and $\eta = 2.5 \times \epsilon/\#\text{steps}$. For standard AT and MART we perform the inner maximization PGD using $\mathcal{L}_{\text{PGD}} = \mathcal{L}_{\text{CE}}(f_{\boldsymbol{\theta}}(\tilde{\mathbf{x}}), y)$, whereas for TRADES we use $\mathcal{L}_{\text{PGD}} = D_{\text{KL}}(f_{\boldsymbol{\theta}}(\tilde{\mathbf{x}})||f_{\boldsymbol{\theta}}(\mathbf{x}))$ as the inner maximization loss, following the original works (Madry et al., 2018; Zhang et al., 2019a; Wang et al., 2020). We set the trade-off parameter $\lambda_{\text{MART}} = 4$ in all experiments, and $\lambda_{\text{TRADES}} = 6$ in CIFAR-10 and $\lambda_{\text{TRADES}} = 3$ in CIFAR-100 and SVHN experiments. We present extended robustness evaluations of the baseline ANNs used in our experiments in Table A1.

**Converted SNNs:** During initialization of layer-wise firing thresholds, we use 10 training mini-batches of size 64 with a calibration sequence length of $T_c = 100$ timesteps, and observe the pre-activation values (Rathi & Roy, 2021; Limbacher et al., 2023). We choose the maximum value in the $\rho = 99.7\%$ percentile of the distribution of observed values. This process is performed similarly to (Lu & Sengupta, 2020), and is a better alternative to simply using the largest observed pre-activation value for threshold balancing (Sengupta et al., 2019), since the procedure involves the use of randomly sampled training batches which might lead to large valued outliers compared to the rest of the distribution. We use $\kappa = 0.3$ in CIFAR-10 and SVHN, and $\kappa = 0.1$ in CIFAR-100 and TinyImageNet experiments. During BPTT-based finetuning, we used the piecewise linear surrogate gradient in Eq (16) with $\gamma_w = 1$, as in Ding et al. (2022). For the initial random step of RFGSM, we set $\alpha = \epsilon_1/2$ in CIFAR-10 and SVHN, and $\alpha = \epsilon_1/5$ in CIFAR-100 and TinyImageNet experiments. For CIFAR-10 experiments with AT-$\epsilon_2$ and AT-$\epsilon_3$, we used $\alpha = \epsilon/4$.

End-to-end trained SNNs used for comparisons were initialized statically at $V_{th}^l = 1$ for all layers, as in Ding et al. (2022). During robust finetuning, we impose a lower bound for the trainable firing thresholds such that they remain $V_{th}^l \geq 0.03$. At inference-time with our models, for tdBN layers, we use the mean and variance moving averages estimated during the robust finetuning process from scratch. Since we performed conversion with baseline ANNs without bias terms (Sengupta et al., 2019; Rueckauer et al., 2016), only in CIFAR-10/100 experiments we introduce bias terms during finetuning in the output dense layers to be also optimized, in order to have identical networks to SNN-RAT. Although it slightly increased model performance, we did not find the use of bias terms strictly necessary in our method.

**Optimization:** We used momentum SGD with a cosine annealing based learning rate scheduler, and a batch size of 64 in all model training settings. All end-to-end trained models (i.e., baseline ANNs and vanilla SNNs) were optimized for 200 epochs with an initial learning rate of 0.1, following the conventional settings (Madry et al., 2018; Zhang et al., 2019a; Wang et al., 2020). After ANN-to-SNN conversion and

Table A1: Robustness evaluations (%) of used baseline ANN architectures in our experiments. All attacks are evaluated under an $l_\infty$-norm bounded $\epsilon_3 = 8/255$ perturbation budget. $AA_\infty$ indicates AutoAttack evaluations. PGD attack step sizes were determined by $2.5 \times \epsilon/\#\text{steps}$. Maximum number of queries for SquareAttack were set to 5000.

| | | Clean Acc. | $\epsilon_3$-Robust Accuracies | | | | | | |
| | | | FGSM | $PGD^{10}$ | $PGD^{20}$ | $PGD^{50}$ | $APGD_{CE}$ | Square | $AA_\infty$ |
|---|---|---|---|---|---|---|---|---|---|
| CIFAR-10 with VGG11 | Natural | 95.10 | 20.35 | 0.13 | 0.09 | 0.06 | 1.07 | 0.45 | 0.00 |
| | AT-$\epsilon_1$ | 93.55 | 42.71 | 21.86 | 21.15 | 20.76 | 18.98 | 32.99 | 16.64 |
| | TRADES-$\epsilon_1$ | 92.10 | 51.26 | 33.59 | 32.94 | 32.56 | 29.75 | 41.90 | 27.62 |
| | MART-$\epsilon_1$ | 92.68 | 47.21 | 28.43 | 27.54 | 27.43 | 24.58 | 37.56 | 21.98 |
| | AT-$\epsilon_2$ | 91.05 | 51.55 | 37.00 | 36.56 | 36.32 | 34.62 | 45.87 | 32.61 |
| | AT-$\epsilon_3$ | 85.07 | 58.30 | 53.57 | 53.38 | 53.24 | 51.65 | 52.98 | 45.05 |
| CIFAR-100 with WRN-16-4 | Natural | 78.93 | 6.79 | 0.00 | 0.00 | 0.00 | 0.00 | 0.00 | 0.00 |
| | AT-$\epsilon_1$ | 74.90 | 20.23 | 9.85 | 9.46 | 9.34 | 8.49 | 15.53 | 7.05 |
| | TRADES-$\epsilon_1$ | 73.87 | 20.40 | 11.07 | 10.74 | 10.53 | 9.64 | 15.61 | 8.43 |
| | MART-$\epsilon_1$ | 72.61 | 23.93 | 15.08 | 14.72 | 14.47 | 13.26 | 19.51 | 11.04 |
| SVHN with ResNet-20 | Natural | 97.08 | 63.83 | 0.71 | 0.15 | 0.08 | 1.87 | 1.38 | 0.08 |
| | AT-$\epsilon_1$ | 96.50 | 52.53 | 30.36 | 29.56 | 29.25 | 25.86 | 29.30 | 23.58 |
| | TRADES-$\epsilon_1$ | 96.29 | 54.11 | 33.43 | 32.63 | 32.18 | 29.13 | 32.10 | 27.16 |
| | MART-$\epsilon_1$ | 96.16 | 56.66 | 36.02 | 35.32 | 34.82 | 31.04 | 31.70 | 27.65 |

setting initial thresholds, SNNs were finetuned for 60 epochs on CIFAR-10, SVHN and TinyImageNet, and 80 epochs on CIFAR-100, with an initial learning rate of 0.001. During robust finetuning, we used a weight decay parameter of 0.001 in CIFAR-10, SVHN and TinyImageNet experiments, whereas we used 0.0001 for CIFAR-100. We also used lower weight decay parameters of 0.0005 and 0.0001 during post-conversion finetuning of AT-$\epsilon_2$ and AT-$\epsilon_3$ models respectively in our CIFAR-10 experiments.

**Computational Overhead:** Although our method uses a larger amount of total training epochs (baseline ANN training and robust SNN finetuning), the overhead does not differ much from end-to-end AT methods (e.g., SNN-RAT) due to the heavier computational load of AT with SNNs which involves the temporal dimension. To illustrate quantitatively (CIFAR-10 with VGG11), Ours: baseline ANN training for 200 epochs takes ~11 hours and robust SNN finetuning for 60 epochs takes ~10.5 hours, SNN-RAT: robust end-to-end training for 200 epochs takes ~23 hours wall-clock time.

**Implementations:** All models were implemented with the PyTorch 1.13.0 (Paszke et al., 2019) library, and experiments were performed using GPU hardware of types NVIDIA A40, NVIDIA Quadro RTX 8000 and NVIDIA Quadro P6000. Adversarial attacks were implemented using the TorchAttacks (Kim, 2020) library, with the default attack hyper-parameters. Our implementations can be found at: https://github.com/IGITUGraz/RobustSNNConversion.

## A.3 Robust SNN Finetuning Parameter Updates with Surrogate Gradients

Our robust optimization objective after the initial ANN-to-SNN conversion step follows:

$$\min_{\boldsymbol{\theta}} \mathbb{E}\left[\mathcal{L}_{CE}(f_{\boldsymbol{\theta}}(\mathbf{x}), y) + \beta \cdot \max_{\tilde{\mathbf{x}} \in \Delta_\epsilon^p(\mathbf{x})} D_{KL}(f_{\boldsymbol{\theta}}(\tilde{\mathbf{x}})||f_{\boldsymbol{\theta}}(\mathbf{x}))\right], \tag{11}$$

where $\boldsymbol{\theta}$ consist of $\{\mathbf{W}^l, \varphi^l, \omega^l\}_{l=1}^L$ and $\{V_{th}^l\}_{l=1}^{L-1}$, the adversarial example $\tilde{\mathbf{x}} \in \Delta_\epsilon^\infty(\mathbf{x})$ is obtained at the inner maximization via RFGSM using $\mathcal{L}_{RFGSM} = D_{KL}(f_{\boldsymbol{\theta}}(\mathbf{x}')||f_{\boldsymbol{\theta}}(\mathbf{x}))$,

$$\mathcal{L}_{CE} = -\sum_j y^{(j)} \log(f_{\boldsymbol{\theta}}^{(j)}(\mathbf{x})) \quad \text{and} \quad D_{KL} = \sum_j f_{\boldsymbol{\theta}}^{(j)}(\tilde{\mathbf{x}}) \log \frac{f_{\boldsymbol{\theta}}^{(j)}(\tilde{\mathbf{x}})}{f_{\boldsymbol{\theta}}^{(j)}(\mathbf{x})}. \tag{12}$$

Following the robust optimization objective that minimizes an overall loss function expressed in the brackets of Eq. (11), say $\mathcal{L}_{\text{robust}}$, the gradient-based synaptic connectivity weight and firing threshold updates in the hidden layers of the SNN $l = \{1, \ldots, L-1\}$ are determined via BPTT as:

$$\Delta \mathbf{W}^l = \sum_t \frac{\partial \mathcal{L}_{\text{robust}}}{\partial \mathbf{W}^l} = \sum_t \frac{\partial \mathcal{L}_{\text{robust}}}{\partial \mathbf{s}^l(t)} \frac{\partial \mathbf{s}^l(t)}{\partial \mathbf{z}^l(t)} \frac{\partial \mathbf{z}^l(t)}{\partial \mathbf{v}^l(t^-)} \frac{\partial \mathbf{v}^l(t^-)}{\partial \mathbf{W}^l}, \tag{13}$$

$$\Delta V_{th}^l = \sum_t \frac{\partial \mathcal{L}_{\text{robust}}}{\partial V_{th}^l} = \sum_t \frac{\partial \mathcal{L}_{\text{robust}}}{\partial \mathbf{s}^l(t)} \frac{\partial \mathbf{s}^l(t)}{\partial \mathbf{z}^l(t)} \frac{\partial \mathbf{z}^l(t)}{\partial V_{th}^l}, \tag{14}$$

where $\mathbf{z}^l(t) = \mathbf{v}^l(t^-) - V_{th}^l$ is the input to the Heaviside step function: $\mathbf{s}^l(t) = H(\mathbf{z}^l(t))$. For BPTT we use the piecewise linear surrogate gradient with a unit dampening factor:

$$\frac{\partial \mathbf{s}^l(t)}{\partial \mathbf{z}^l(t)} := \max\{0, 1 - |\mathbf{z}^l(t)|\}. \tag{15}$$

Remaining gradient terms can be computed via LIF neuron dynamics. For the layers containing tdBN operations, the parameters $\varphi^l$ and $\omega^l$ are also updated similarly through normalized $\hat{\mathbf{s}}^l(t)$, following the original gradient computations from Ioffe & Szegedy (2015), which are however gathered temporally (Zheng et al., 2021). Since there is no spiking activity in the output layer, weight updates can be computed via BPTT without surrogate gradients, similar to the definitions from (Kundu et al., 2021; Rathi & Roy, 2021). Our complete robust ANN-to-SNN conversion procedure is outlined in Algorithm 1.

## A.4 Ensemble Adversarial Attacks

The success of an SNN attack depends critically on the surrogate gradient used for by the adversary. Therefore we consider adaptive adversaries that perform a number of attacks for any test sample, where the surrogate gradient used to approximate the discontinuity of the spike function during BPTT varies in each case. Specifically, as part of the attack ensemble we perform backpropagation by using the piecewise linear (triangular) function (Bellec et al., 2018; Neftci et al., 2019) defined as in Eq. (16) with $\gamma_w \in \{0.25, 0.5, 1.0, 2.0, 3.0\}$,

$$\frac{\partial \mathbf{s}^l(t)}{\partial \mathbf{z}^l(t)} := \frac{1}{\gamma_w^2} \cdot \max\{0, \gamma_w - |\mathbf{z}^l(t)|\}, \tag{16}$$

the exponential surrogate gradient function (Shrestha & Orchard, 2018) defined as in Eq. (17) with hyperparameters $(\gamma_d, \gamma_s) \in \{(0.3, 0.5), (0.3, 1.0), (0.3, 2.0), (1.0, 0.5), (1.0, 1.0), (1.0, 2.0)\}$,

$$\frac{\partial \mathbf{s}^l(t)}{\partial \mathbf{z}^l(t)} := \gamma_d \cdot \exp\left(-\gamma_s \cdot |\mathbf{z}^l(t)|\right), \tag{17}$$

and the rectangular function (Wu et al., 2018) defined as in Eq. (18) with $\gamma_w \in \{0.25, 0.5, 1.0, 2.0, 4.0\}$,

$$\frac{\partial \mathbf{s}^l(t)}{\partial \mathbf{z}^l(t)} := \frac{1}{\gamma_w} \cdot \text{sign}\left(|\mathbf{z}^l(t)| < \frac{\gamma_w}{2}\right). \tag{18}$$

For the rectangular and piecewise linear (triangular) functions, $\gamma_w$ indicates the width that determines the input range to activate the gradient. The dampening factor of these surrogate gradients is then inversely proportional to $\gamma_w$ (Ding et al., 2022; Deng et al., 2022; Wu et al., 2018; 2019). For the exponential function, $\gamma_d$ controls the dampening and $\gamma_s$ controls the steepness. A conventional choice for the exponential is $(\gamma_d, \gamma_s) = (0.3, 1.0)$ (Shrestha & Orchard, 2018), while we also experimented by varying these parameters. Interestingly, we could verify that varying the shape and parameters of this surrogate gradient function occasionally had an impact on the success of the adversary for various test samples. This scenario, for instance, indicates having approximate gradient information long before the membrane potential reaches the firing threshold (i.e., larger $\gamma_w$), which can be useful to attack SNNs from the perspective of an adaptive adversary. In Section B.2 we present the effectiveness of the individual components of the ensemble.

---

**Algorithm 1** Adversarially robust ANN-to-SNN conversion

---

1: **Input:** Training dataset $\mathcal{D}$, adversarially pre-trained baseline ANN parameters $\{\mathbf{W}^l, \varphi^l, \omega^l\}_{l=1}^L$, number of calibration samples, calibration sequence length $T_c$, percentile $\rho$, threshold scaling factor $\kappa$, number of finetuning iterations, perturbation strength $\epsilon$ and random step $\alpha$ for RFGSM, simulation timesteps $T$, membrane leak factor $\tau$, trade-off parameter $\beta$.

2: **Output:** Robust SNN $f$ with parameters $\boldsymbol{\theta}$ consisting of $\{\mathbf{W}^l, \varphi^l, \omega^l\}_{l=1}^L$ and $\{V_{th}^l\}_{l=1}^{L-1}$.

    ▷ Converting the adversarially trained baseline ANN

3: **Initialize:** Set weights of the SNN $f$ directly from the pre-trained ANN parameters $\{\mathbf{W}^l, \varphi^l, \omega^l\}_{l=1}^L$.

4: **for** $l = 1$ **to** $L - 1$ **do**

5:     **for** $c = 1$ **to** $\#calibration\_samples$ **do**

6:        **for** $t = 1$ **to** $T_c$ **do**

7:           $\mathbf{r}^l \leftarrow$ Store pre-activation values at layer $l$ during forward pass with direct input coding.

8:        **end for**

9:        **if** $\max[\rho$-percentile of the dist. in $\mathbf{r}^l] > V_{th}^l$ **then**

10:           $V_{th}^l = \max[\rho$-percentile of the distribution in $\mathbf{r}^l]$

11:        **end if**

12:     **end for**

13: **end for**

    ▷ Initializing trainable firing thresholds

14: **for** $l = 1$ **to** $L - 1$ **do**

15:     $V_{th}^l \leftarrow \kappa \cdot V_{th}^l$

16: **end for**

    ▷ Robust finetuning of SNN after conversion

17: **for** $i = 1$ **to** $\#finetuning\_iterations$ **do**

18:     Sample a mini-batch of $(\mathbf{x}, y) \sim \mathcal{D}$

       ▷ Inner maximization for single-step adversarial examples

19:     $\mathbf{x}' \leftarrow \mathbf{x} + \alpha \cdot \text{sign}\left(\mathcal{N}(\mathbf{0}, \mathbf{I})\right)$

20:     $f_{\boldsymbol{\theta}}(\mathbf{x}'), f_{\boldsymbol{\theta}}(\mathbf{x}) \leftarrow$ Forward pass direct coded $\mathbf{x}'$ and $\mathbf{x}$ via Eqs. (1), (2), (3), (7), with $\tau$.

21:     $\tilde{\mathbf{x}} \leftarrow \mathbf{x}' + (\epsilon - \alpha) \cdot \text{sign}\left(\nabla_{\mathbf{x}'} \mathcal{L}_{\text{RFGSM}}\right)$ with $\mathcal{L}_{\text{RFGSM}} = D_{\text{KL}}(f_{\boldsymbol{\theta}}(\mathbf{x}') || f_{\boldsymbol{\theta}}(\mathbf{x}))$.

       ▷ Outer optimization with the robust finetuning objective

22:     $f_{\boldsymbol{\theta}}(\mathbf{x}), f_{\boldsymbol{\theta}}(\tilde{\mathbf{x}}) \leftarrow$ Forward pass direct coded $\mathbf{x}$ and $\tilde{\mathbf{x}}$ via Eqs. (1), (2), (3), (7), with $\tau$.

23:     $\mathcal{L}_{\text{robust}} \leftarrow \mathbb{E}\left[\mathcal{L}_{\text{CE}}(f_{\boldsymbol{\theta}}(\mathbf{x}), y) + \beta \cdot D_{\text{KL}}\left(f_{\boldsymbol{\theta}}(\tilde{\mathbf{x}}) || f_{\boldsymbol{\theta}}(\mathbf{x})\right)\right]$

24:     $\Delta \mathbf{W}^l \leftarrow \sum_t \frac{\partial \mathcal{L}_{\text{robust}}}{\partial \mathbf{W}^l}$

25:     $\Delta \varphi^l \leftarrow \sum_t \frac{\partial \mathcal{L}_{\text{robust}}}{\partial \varphi^l}$

26:     $\Delta \omega^l \leftarrow \sum_t \frac{\partial \mathcal{L}_{\text{robust}}}{\partial \omega^l}$

27:     $\Delta V_{th}^l \leftarrow \sum_t \frac{\partial \mathcal{L}_{\text{robust}}}{\partial V_{th}^l}$

28: **end for**

---

In the ensemble, we also considered the straight-through estimator (STE) (Bengio et al., 2013), backward pass through rate (BPTR) (Ding et al., 2022), and a conversion-based approximation, where spiking neurons are replaced with ReLU functions during BPTT (Ding et al., 2022). STE uses an identity function as the surrogate gradient for all SNN neurons during backpropagation. BPTR performs a differentiable approximation by taking the derivative of the spike functions directly from the average firing rate of the neurons between layers (Ding et al., 2022). In this case the gradient does not accumulate through time, and the complete neuronal dynamic is approximated during backpropagation by a single STE that maintains gradients for neurons that have a non-zero spike rate in the forward pass. We use the same BPTR implementation based on the evaluations from Ding et al. (2022). Conversion-based approximation was one of the earliest attempts to design adversarial attacks on SNNs which we also include, although we found it to be only minimally effective within the ensemble (Sharmin et al., 2019).

Table B1: Evaluating $\epsilon_1 = 2/255$ and $\epsilon_3 = 8/255$ robust accuracies under black-box (BB) transfer attacks (Papernot et al., 2017), obtained via adversarial examples crafted with ensemble PGD$^{20}$ attacks on a source model. In white-box (WB) setting, the source and target models are identical. BB$_{\text{SNN}}$ indicates attacks via a source model that undergoes the same ANN-to-SNN conversion procedure as the target model, with a different random seed during finetuning. BB$_{\text{ANN}}$ indicates transfer attacks where the source model is the baseline ANN used for conversion (hence no ensemble needed).

| | | Clean Acc. | Adversarially Robust ANN-to-SNN Conversion | | | | | |
| | | | $\epsilon_1$-Robust Accuracies | | | $\epsilon_3$-Robust Accuracies | | |
| | | | WB | BB$_{\text{SNN}}$ | BB$_{\text{ANN}}$ | WB | BB$_{\text{SNN}}$ | BB$_{\text{ANN}}$ |
|---|---|---|---|---|---|---|---|---|
| CIFAR-10 with VGG11 | Natural | 93.53 | 59.22 | 76.93 | 82.21 | 0.74 | 6.87 | 21.99 |
| | **AT-$\epsilon_1$** | 92.14 | 70.02 | 85.40 | 82.04 | 13.18 | 42.27 | 29.81 |
| | **TRADES-$\epsilon_1$** | 91.39 | 72.00 | 86.21 | 82.29 | 19.65 | 52.34 | 38.18 |
| | **MART-$\epsilon_1$** | 91.47 | 70.35 | 84.19 | 81.43 | 17.79 | 41.82 | 32.26 |
| CIFAR-100 with WRN-16-4 | Natural | 70.87 | 35.49 | 45.95 | 64.44 | 1.40 | 3.15 | 48.12 |
| | **AT-$\epsilon_1$** | 70.02 | 44.73 | 54.88 | 59.36 | 7.64 | 11.88 | 25.52 |
| | **TRADES-$\epsilon_1$** | 67.84 | 42.73 | 53.14 | 58.20 | 7.79 | 11.86 | 27.23 |
| | **MART-$\epsilon_1$** | 67.10 | 43.46 | 53.21 | 57.83 | 8.90 | 13.39 | 28.03 |
| SVHN with ResNet-20 | Natural | 94.18 | 74.30 | 82.21 | 91.71 | 10.35 | 14.96 | 89.18 |
| | **AT-$\epsilon_1$** | 91.90 | 71.80 | 81.47 | 84.18 | 21.51 | 29.94 | 44.51 |
| | **TRADES-$\epsilon_1$** | 92.78 | 74.27 | 83.30 | 85.00 | 23.96 | 31.62 | 43.37 |
| | **MART-$\epsilon_1$** | 92.71 | 74.82 | 82.94 | 85.67 | 24.11 | 33.01 | 44.27 |

**Attacks on the Poisson encoding BNTT model:** In our TinyImageNet experiments, for FGSM and PGD$^{20}$ on the BNTT model (Kim & Panda, 2021b), we used the SNN-crafted BPTT-based adversarial input generation methodology by Sharmin et al. (2020). These attacks approximate the non-differentiable Poisson input encoding layer by using the gradients obtained for the first convolution layer activations during the attack, and was found to be a powerful substitute. We also used two expectation-over-transformation (EOT) iterations (Athalye et al., 2018b) while crafting adversarial examples on this model, due to the involved randomness in input encoding (Athalye et al., 2018a). Overall, we use the same surrogate gradient ensemble approach, however with an obligatory approximation of the input encoding and EOT iterations for stable gradient estimates.

# B   Additional Experiments

## B.1   Black-box Transfer Attacks

In Table B1 we present our transfer-based black-box robustness evaluations (Papernot et al., 2017), using both the baseline ANN, and a similar SNN model obtained with the same conversion procedure using a different random seed. We craft adversarial examples on the source models with PGD$^{20}$ also in an ensemble setting. In all models, we observe that white-box robust accuracies are significantly lower than black-box cases. This result holds both for the $\epsilon_1$ perturbation budget where the models were trained and finetuned on, as well as $\epsilon_3$ where we investigate robust generalization to higher perturbation bounds. Only in CIFAR-10, we observed that adversarially trained baseline ANNs used for conversion were able to craft stronger adversarial examples than the SNNs converted and finetuned in an identical setting (e.g., $\epsilon_3$-robustness for AT-$\epsilon_1$: BB$_{\text{ANN}}$: 29.81%, BB$_{\text{SNN}}$: 42.27%). We believe that this is related to our weaker choice of $\beta = 2$ for finetuning in CIFAR-10 experiments, whereas we used $\beta = 4$ in all other models. On another note, we observed that BB$_{\text{SNN}}$ attacks are highly powerful on converted SNNs based on natural baseline ANN training (e.g., CIFAR-10 $\epsilon_3$-robustness WB: 0.74%, BB$_{\text{SNN}}$: 6.87%), demonstrating their weakness even in the black-box setting.

Table B2: Evaluating $\epsilon_3 = 8/255$ FGSM attack robust accuracies from our CIFAR-10 experiments with VGG11, under individual components of the white-box ensemble. Triangular (piecewise linear), exponential and rectangular surrogate gradients were used as part of BPTT. Underlined values indicate the strongest individual attack within the ensemble.

| | | SNN-RAT | ANN-to-SNN Conversion (Ours) | | | |
|---|---|---|---|---|---|---|
| | | | Natural | AT-$\epsilon_1$ | TRADES-$\epsilon_1$ | MART-$\epsilon_1$ |
| | Clean Accuracy | 90.74 | 93.53 | 92.14 | 91.39 | 91.47 |
| | Ensemble FGSM | **32.80** | **15.71** | **33.94** | **41.42** | **44.39** |
| Triangular | $\gamma_w = 1.0$ | 45.01 | 28.16 | 53.61 | 71.50 | 62.16 |
| | $\gamma_w = 2.0$ | 39.27 | 63.06 | 74.36 | 82.94 | 79.27 |
| | $\gamma_w = 3.0$ | 43.23 | 77.79 | 83.59 | 87.72 | 86.30 |
| | $\gamma_w = 0.5$ | 82.20 | 19.62 | 39.25 | 52.09 | 51.56 |
| | $\gamma_w = 0.25$ | 90.15 | 54.15 | 57.48 | 51.03 | 50.16 |
| Exponential | $(\gamma_d, \gamma_s) = (0.3, 0.5)$ | 48.94 | 80.75 | 83.89 | 87.68 | 86.70 |
| | $(\gamma_d, \gamma_s) = (0.3, 1.0)$ | 36.91 | 54.96 | 69.65 | 79.66 | 74.97 |
| | $(\gamma_d, \gamma_s) = (0.3, 2.0)$ | 37.65 | 27.15 | 51.66 | 67.60 | 60.02 |
| | $(\gamma_d, \gamma_s) = (1.0, 0.5)$ | 62.73 | 89.41 | 89.01 | 90.36 | 89.41 |
| | $(\gamma_d, \gamma_s) = (1.0, 1.0)$ | 41.63 | 64.45 | 75.37 | 83.30 | 79.34 |
| | $(\gamma_d, \gamma_s) = (1.0, 2.0)$ | 38.65 | 32.07 | 55.96 | 70.99 | 62.55 |
| Rectangular | $\gamma_w = 0.25$ | 90.01 | 88.14 | 85.22 | 71.08 | 66.57 |
| | $\gamma_w = 0.5$ | 89.15 | 26.58 | 42.51 | 51.18 | 53.44 |
| | $\gamma_w = 1.0$ | 65.28 | 25.10 | 49.85 | 68.98 | 61.68 |
| | $\gamma_w = 2.0$ | 45.21 | 63.45 | 73.59 | 83.49 | 80.22 |
| | $\gamma_w = 4.0$ | 52.62 | 88.84 | 89.42 | 90.93 | 89.43 |
| | Backward Pass Through Rate | 51.46 | 23.90 | 41.33 | 46.68 | 58.26 |
| | Conversion-based Approx. | 84.25 | 39.73 | 55.35 | 89.43 | 87.79 |
| | Straight-Through Estimation | 90.14 | 92.84 | 90.74 | 90.68 | 91.03 |

**On the Analysis of Gradient Obfuscation:** Previous SNN robustness studies (Kundu et al., 2021; Ding et al., 2022) analyzes gradient obfuscation (Athalye et al., 2018a) solely based on the five principles: (1) iterative attacks should perform better than single-step attacks, (2) white-box attacks should perform better than transfer-based attacks, (3) increasing attack perturbation bound should result in lower robust accuracies, (4) unbounded attacks can nearly reach 100% success[3], (5) adversarial examples can not be found through random sampling[4]. In the main paper and our results presented here, we could confirm all of these observations in our evaluations. However, we argue that more reliable SNN assessments should ideally go beyond these, as we clarify in further depth via our ensemble evaluations (Carlini et al., 2019).

## B.2    Evaluations with Alternative Surrogate Gradient Functions

Our detailed results for one particular ensemble attack is demonstrated in Table B2, where we focus on single-step FGSM without an initial randomized start. The third row indicates a naive BPTT-based FGSM attack where the adversary uses the same surrogate gradient function utilized during model training (e.g., SNN-RAT: 45.01%, Ours (AT-$\epsilon_1$): 53.61%). Importantly, we observe that there are stronger attacks that can be crafted by a versatile adversary, by varying the shape or parameters of the used surrogate gradient. Moreover, in this particular case with FGSM attacks, we observe 4-6% difference in attack performance between the strongest individual attack configuration within the ensemble (e.g., underlined SNN-RAT: 36.91%, Ours (AT-$\epsilon_1$): 39.25%), versus using an ensemble of alternative possibilities (e.g., SNN-RAT: 32.80%,

---

[3]After a perturbation bound of 16/255, ensemble attacks on all models reached $< 1\%$ robust accuracy.
[4]Gaussian noise with $\epsilon_3$ standard deviation only led to a 0.4-3.0% decrease from the clean acc. of the models.

Ours (AT-$\epsilon_1$): 33.94%). This further emphasizes the relative strength of our evaluations, in comparison to traditional naive evaluation settings.

Table B3 shows a comparison of ANNs and SNNs under *analogous* PGD$^{20}$ attack settings. Here, SNN evaluations under PGD$^{20}_{\mathrm{vanilla}}$ indicate naive BPTT-based attacks, where the adversary only uses the same surrogate gradient utilized during training. Simply running these evaluations would generally be considered analogous to those for the PGD$^{20}$ ANN attacks (Ding et al., 2022), although they are essentially not identical. Considering PGD$^{20}_{\mathrm{vanilla}}$ accuracies, results might even

Table B3: Impact of ensemble SNN attacks on CIFAR-10/100, with comparisons to ANNs under the analogous $\epsilon_3$-attack. PGD$^{20}_{\mathrm{vanilla}}$ uses BPTT with training-time surrogate gradient.

| CIFAR-10 with VGG11 | ANN | | SNN | | |
|---|---|---|---|---|---|
| | Clean | PGD$^{20}$ | Clean | PGD$^{20}_{\mathrm{vanilla}}$ | PGD$^{20}_{\mathrm{ensemble}}$ |
| **AT-$\epsilon_1$** | 93.55 | 21.15 | 92.14 | 37.42 | 13.18 |
| **TRADES-$\epsilon_1$** | 92.10 | 32.94 | 91.39 | 62.03 | 19.65 |
| **MART-$\epsilon_1$** | 92.68 | 27.54 | 91.47 | 49.90 | 17.79 |

| CIFAR-100 with WRN-16-4 | ANN | | SNN | | |
|---|---|---|---|---|---|
| | Clean | PGD$^{20}$ | Clean | PGD$^{20}_{\mathrm{vanilla}}$ | PGD$^{20}_{\mathrm{ensemble}}$ |
| **AT-$\epsilon_1$** | 74.90 | 9.46 | 70.02 | 12.56 | 7.64 |
| **TRADES-$\epsilon_1$** | 73.87 | 10.74 | 67.84 | 12.97 | 7.79 |
| **MART-$\epsilon_1$** | 72.61 | 14.72 | 67.10 | 14.27 | 8.90 |

be (wrongly) perceived as SNNs being more robust than ANNs, as claimed in previous work based on shallow models and naive evaluation settings (Kim & Panda, 2021c; Li et al., 2023). However, since the SNN attack depends on the surrogate gradient, we also consider the ensemble PGD$^{20}$, as we did throughout our work. Note that this stronger attack is not necessarily equivalent to the ANN attack either, but we believe it is a better basis for comparison.

**Comparison to RGA Attacks:** We compare our ensemble attack approach also against a recently proposed, concurrent work on rate gradient approximation (RGA) attacks on SNNs (Bu et al., 2023). Our results demonstrated in Table B4 show that the proposed ensemble approach yields significantly stronger robust accuracy estimates both in our CIFAR-10 and CIFAR-100 simulations, and presents an already significantly stronger evaluation baseline than RGA for the main models considered in this work (e.g., for experiments CIFAR-10 with VGG11, SNN-RAT under PGD$^{20}_{\mathrm{ensemble}}$: 10.06% vs. PGD$^{20}_{\mathrm{RGA}}$: 17.98%, or Ours (TRADES-$\epsilon_1$) under PGD$^{20}_{\mathrm{ensemble}}$: 19.65% vs. PGD$^{20}_{\mathrm{RGA}}$: 26.51%). Nevertheless, it is indeed possible to

Table B4: Comparison of ensemble SNN attacks against RGA attacks (Bu et al., 2023) under $\epsilon_3$ with PGD$^{20}$.

| CIFAR-10 with VGG11 | Clean | PGD$^{20}_{\mathrm{RGA}}$ | PGD$^{20}_{\mathrm{ensemble}}$ |
|---|---|---|---|
| SNN-RAT | 90.74 | 17.98 | **10.06** |
| Ours (Natural) | 93.53 | 1.50 | **0.74** |
| Ours (AT-$\epsilon_1$) | 92.14 | 16.44 | **13.18** |
| Ours (TRADES-$\epsilon_1$) | 91.39 | 26.51 | **19.65** |
| Ours (MART-$\epsilon_1$) | 91.47 | 24.53 | **17.79** |

| CIFAR-100 with WRN-16-4 | Clean | PGD$^{20}_{\mathrm{RGA}}$ | PGD$^{20}_{\mathrm{ensemble}}$ |
|---|---|---|---|
| SNN-RAT | 69.32 | 7.72 | **6.04** |
| Ours (Natural) | 70.87 | 5.50 | **1.40** |
| Ours (AT-$\epsilon_1$) | 70.02 | 16.81 | **7.64** |
| Ours (TRADES-$\epsilon_1$) | 67.84 | 16.16 | **7.79** |
| Ours (MART-$\epsilon_1$) | 67.10 | 17.43 | **8.90** |

integrate RGA (Bu et al., 2023) as an alternative method to approximate more stable gradients for the adversary, within our proposed ensemble SNN attack framework in future work.

## B.3   Ablations on the Robust SNN Finetuning Objective

Our ablation studies are presented in Table B5. The first three rows respectively denote comparisons to end-to-end trained SNNs using a natural cross-entropy loss, and using our robust finetuning objective without and with trainable thresholds, for 200 epochs. Note that in the third row, we are essentially comparing our ANN-to-SNN conversion approach, to simply end-to-end adversarially training the SNN using the TRADES objective. Our method yields approximately twice as more robust models than the model in the third row (i.e., ensemble APGD$_{\mathrm{CE}}$: 4.72 versus 10.05), indicating that the weight initialization from an adversarially trained ANN brings significant gains. The fourth model indicates weight initialization from a naturally trained ANN within our conversion approach (previously included in Table 1 as well), which yields nearly no robustness since these weights are inclined to a non-robust configuration during natural ANN training.

Remaining models compare different choices for $\mathcal{L}_{\mathrm{RFGSM}}$ (e.g., adversarial cross-entropy loss (Tramèr et al., 2018)), and the outer optimization scheme (e.g., standard AT (Madry et al., 2018)), which we have eventually

Table B5: Ablations for the robust SNN finetuning objective with our CIFAR-10 experiments. Robust accuracies are evaluated via ensemble white-box attacks. Bottom row indicates our method.

| Baseline ANN | Inner Max. $\mathcal{L}_{\mathrm{RFGSM}}$ | Outer Optimization | Trainable $V_{th}^l$ | Clean Acc. | $\epsilon_3$-Robust Accuracies | | |
|:---:|:---:|:---:|:---:|:---:|:---:|:---:|:---:|
| | | | | | FGSM | PGD$^{20}$ | APGD$_{\mathrm{CE}}$ |
| ✗ | − | $\mathcal{L}_{\mathrm{CE}}(f_\theta(\mathbf{x}),y)$ | ✗ | 93.73 | 6.43 | 0.00 | 0.00 |
| ✗ | $D_{\mathrm{KL}}(f_\theta(\mathbf{x}')\|f_\theta(\mathbf{x}))$ | Eq. (11) | ✗ | 91.90 | 24.94 | 7.23 | 5.32 |
| ✗ | $D_{\mathrm{KL}}(f_\theta(\mathbf{x}')\|f_\theta(\mathbf{x}))$ | Eq. (11) | ✓ | 91.75 | 23.83 | 6.70 | 4.72 |
| Natural | $D_{\mathrm{KL}}(f_\theta(\mathbf{x}')\|f_\theta(\mathbf{x}))$ | Eq. (11) | ✓ | 93.53 | 15.71 | 0.74 | 0.44 |
| **AT-$\epsilon_1$** | − | $\mathcal{L}_{\mathrm{CE}}(f_\theta(\mathbf{x}),y)$ | ✓ | 92.18 | 32.31 | 10.03 | 8.40 |
| **AT-$\epsilon_1$** | $\mathcal{L}_{\mathrm{CE}}(f_\theta(\mathbf{x}'),y)$ | $\mathcal{L}_{\mathrm{CE}}(f_\theta(\tilde{\mathbf{x}}),y)$ | ✓ | 92.02 | 34.17 | 12.19 | 9.40 |
| **AT-$\epsilon_1$** | $D_{\mathrm{KL}}(f_\theta(\mathbf{x}')\|f_\theta(\mathbf{x}))$ | $\mathcal{L}_{\mathrm{CE}}(f_\theta(\tilde{\mathbf{x}}),y)$ | ✓ | 91.59 | 32.83 | 11.78 | 9.26 |
| **AT-$\epsilon_1$** | $\mathcal{L}_{\mathrm{CE}}(f_\theta(\mathbf{x}'),y)$ | Eq. (11) | ✓ | 92.13 | 33.15 | 11.96 | 9.32 |
| **AT-$\epsilon_1$** | $D_{\mathrm{KL}}(f_\theta(\mathbf{x}')\|f_\theta(\mathbf{x}))$ | Eq. (11) | ✗ | 91.09 | 34.03 | 11.51 | 9.39 |
| **AT-$\epsilon_1$ (Ours)** | $D_{\mathrm{KL}}(f_\theta(\mathbf{x}')\|f_\theta(\mathbf{x}))$ | Eq. (11) | ✓ | **92.14** | **33.94** | **13.18** | **10.05** |

Table B6: Evaluations of vanilla and adversarially robust SNNs on out-of-distribution generalization to common image corruptions and perturbations using the CIFAR-10-C test set (Hendrycks & Dietterich, 2019). Values indicate averaged accuracies across the complete test set for all five severities.

| | Clean Acc. | C10-C (all) | C10-C (w/o noise) | Noise | | | Blur | | | | Weather | | | | Digital | | | |
|---|---|---|---|---|---|---|---|---|---|---|---|---|---|---|---|---|---|---|
| | | | | Gauss | Shot | Impulse | Defocus | Glass | Motion | Zoom | Snow | Frost | Fog | Bright | Contrast | Elastic | Pixel | JPEG |
| Vanilla SNN | 93.73 | 81.22 | 83.04 | 72.3 | 79.0 | 70.5 | 85.1 | 73.3 | 80.0 | 83.4 | 85.3 | 85.8 | 80.9 | 91.8 | 67.5 | 86.6 | 88.4 | 88.4 |
| SNN-RAT | 90.74 | 81.72 | 81.60 | 83.4 | 85.9 | 77.3 | 85.2 | 83.3 | 80.8 | 83.8 | 83.5 | 83.2 | 72.3 | 87.7 | 56.1 | 85.3 | 89.2 | 88.7 |
| **Ours (AT-$\epsilon_1$)** | 92.14 | 83.85 | 83.66 | 86.3 | 88.0 | 79.5 | 86.2 | 82.0 | 81.9 | 84.9 | 85.8 | 87.3 | 74.9 | 90.7 | 65.3 | 85.3 | 90.0 | 89.6 |

Table B7: Evaluations of vanilla and adversarially robust SNNs on out-of-distribution generalization to common image corruptions and perturbations using the CIFAR-100-C test set (Hendrycks & Dietterich, 2019). Values indicate averaged accuracies across the complete test set for all five severities.

| | Clean Acc. | C100-C (all) | C100-C (w/o noise) | Noise | | | Blur | | | | Weather | | | | Digital | | | |
|---|---|---|---|---|---|---|---|---|---|---|---|---|---|---|---|---|---|---|
| | | | | Gauss | Shot | Impulse | Defocus | Glass | Motion | Zoom | Snow | Frost | Fog | Bright | Contrast | Elastic | Pixel | JPEG |
| Vanilla SNN | 74.10 | 46.14 | 50.60 | 20.7 | 29.0 | 35.2 | 56.8 | 19.9 | 50.9 | 51.9 | 51.5 | 46.0 | 59.2 | 69.1 | 48.2 | 57.2 | 46.63 | 49.9 |
| SNN-RAT | 69.32 | 51.95 | 54.26 | 39.4 | 45.4 | 43.5 | 58.5 | 40.6 | 52.8 | 56.2 | 57.0 | 54.6 | 47.9 | 64.6 | 35.3 | 59.0 | 61.8 | 62.9 |
| **Ours (AT-$\epsilon_1$)** | 70.02 | 54.06 | 55.54 | 42.5 | 49.3 | 52.5 | 59.6 | 41.2 | 53.6 | 57.5 | 58.6 | 57.5 | 49.2 | 66.0 | 38.4 | 58.9 | 62.0 | 64.1 |

chosen to follow a similar one to TRADES (Zhang et al., 2019a). We also observed significant robustness benefits of using trainable $V_{th}^l$ during post-conversion finetuning as proposed in Rathi & Roy (2021) (e.g., clean/PGD$^{20}$: 91.09/9.39 vs. 92.14/10.05). Overall, these results show that our design choices were important to achieve better robustness. Nevertheless, several alternatives in Table B5 (e.g., standard AT based outer optimization or $\mathcal{L}_{\mathrm{CE}}(f_\theta(\mathbf{x}'),y)$ based RFGSM) still remained effective to outperform existing defenses.

**Robust ANN Conversion Without Finetuning:** In another ablation study, we evaluate our method without the robust finetuning phase altogether. It is important to note that in hybrid conversion, finetuning is a critical component to shorten the simulation length such that one can achieve low-latency. Since we also reset the moving average batch statistics of tdBN layers during conversion, these parameters can only be estimated reliably during the robust finetuning phase. Thus, right after conversion and setting the thresholds, our SNNs only achieve chance-level accuracies with direct input coding for a simulation length of $T = 8$.

However, we evaluated our models (CIFAR-10 with VGG-11) that were only finetuned for a single epoch, or only two epochs to exemplify. After one epoch of robust finetuning the SNN could only achieve a clean/PGD$^{20}$ robust acc. of 87.25%/11.42%, and with only two epochs of robust finetuning it achieves 89.15%/12.97%, whereas our model obtains 92.14%/13.18% in the same setting after 60 epochs of finetuning. These results reveal the benefit of a reasonable robust finetuning stage, both in terms of obtaining more reliable tdBN batch statistics following conversion, as well as adversarial finetuning of layer-wise firing thresholds.

### B.4 Out-of-Distribution Generalization to Common Image Corruptions

It has been argued that adversarial training can have the potential to improve out-of-distribution generalization (Kireev et al., 2022). Tables B6 and B7 presents our evaluations on CIFAR-10-C and CIFAR-100-C respectively (Hendrycks & Dietterich, 2019). Results show that averaged accuracies across all corruptions (noise, blur, weather and digital) improved with our AT-$\epsilon_1$ model (i.e., CIFAR-10-C: 83.85% and CIFAR-100-C: 54.06%), in comparison to SNN-RAT and vanilla SNNs. Importantly, this improvement persists across all individual corruptions, and CIFAR-10/100-C performance without the noise corruptions maintains a similar behavior (see w/o noise averages in Tables B6 and B7). Notably for CIFAR-10-C we observe that SNN-RAT provides noise robustness, but mostly not improving performance in other corruptions (i.e., CIFAR-10-C (all): 81.72%, CIFAR-10-C (w/o noise): 81.60%).

Overall, our conversion-based approach achieved to significantly improve SNNs in terms of their adversarial robustness and generalization. Nevertheless, there are also various other security aspects that were not in the scope of our current work (e.g., preserving privacy in conversion-based SNNs (Kim et al., 2022b)), which requires attention in future studies towards developing trustworthy SNN-based AI applications.

