# OpenReview forum: "Adversarially Robust Spiking Neural Networks Through Conversion"
_TMLR — Accepted by TMLR_

### Review · Reviewer_xfzq · 2024-02-08

**Summary Of Contributions:**

The paper deals with the adversarial robustness of spiking neural networks (SNNs). In particular, they consider the setup where a standard NN is first trained (e.g., a CNN on CIFAR-10) and a spiking SNN is then "distilled" by copying the weight matrices and initialising the other parameters (e.g., the firing rates). The SNN is then fine-tuned on the same dataset (properly encoded).

They show that in order to obtain good results, one has to combine adversarial training of the original NN and adversarial fine-tuning of the resulting SNN. This last step is only done on the firing rate thresholds (the weights are kept fixed) and on the parameters of the batch normalization layers.

They provide an extensive set of results showing the good performance of their method.

**Audience:**

Yes

**Claims And Evidence:**

Yes

**Requested Changes:**

In Section 3.2, I do not follow why the authors are using the RFGSM loss if they described another one previously. In general, it is a bit hard to distinguish what is novel in the paper from what has been lifted from previous literature.

As an example, I would ask a clarification on the novelty of Section "Conversion of Batch Normalization". If I understand correctly, the authors are advocating (i) discarding the previous statistics, (ii) re-computing them (across time and batches) for the SNN, and (iii) fine-tune the trainable weights?

**Strengths And Weaknesses:**

The idea described in the paper is relatively simple, so the paper is easy to follow. I have appreciated the care put into the experimental evaluation in terms of baselines, attacks, etc. Some figures of the proposed method could have improved even more the readability.

As a general comment, I am not sure I understand the motivation of the paper. There is still a sizeable gap in performance between SSNs and standard NNs when applied to these datasets (a gap which is shrinking but still there), so caring about the robustness of these models at this point is a bit strange. The proposed method does not extend to more general "spiking" datasets.

In the experimental part, I noted the lack of clear visualisations of the adversarial attacks. As a sidenote, I am familiar with the literature on adversarial attacks on CNNs but less so on SNNs, so I am not sure about how these visualisations can be achieved. However, I am left wondering whether the resulting adversarial attacks are "easy to spot" by eye.

Overall, this is a technically valid paper. From what I know of the literature, it is novel, although for now it is more of an interesting observation ("SNNs must be initialised from robust NNs to be robust") than a practical concern.

---

> ### Author Response · Authors · 2024-03-05
> **Response to Reviewer xfzq**
>
> >As a general comment, I am not sure I understand the motivation of the paper. There is still a sizeable gap in performance between SSNs and standard NNs when applied to these datasets (a gap which is shrinking but still there), so caring about the robustness of these models at this point is a bit strange. The proposed method does not extend to more general ``spiking" datasets.
>
> We thank the reviewer for raising this comment. We aimed to clarify this early on in our manuscript, in the Introduction section.
> We acknowledge the two main streams of applications for SNNs.  Firstly, they are applied directly to spiking input streams, e.g., from dynamic vision sensors. Secondly, they are also widely applied to standard machine learning tasks and data sets (e.g., image classification). We focus in this article on the latter case. Here, conversion methods have been shown to yield SNN models with excellent performance [Roy et al., 2019], and post-conversion finetuning can lead to low-latency and significantly energy-efficient SNNs [Davies et al., 2021]. In this context, since SNNs offer a promising energy-efficient technology, we argue that their robustness should be an important concern to be studied for reliable embedded AI applications. Since temporal operation characteristics of SNNs make it non-trivial to directly apply existing methods proposed for ANNs, here we develop dedicated algorithms addressing the robustness problem in the SNN domain.
>
> >In the experimental part, I noted the lack of clear visualisations of the adversarial attacks. As a sidenote, I am familiar with the literature on adversarial attacks on CNNs but less so on SNNs, so I am not sure about how these visualisations can be achieved. However, I am left wondering whether the resulting adversarial attacks are "easy to spot" by eye.
>
> We added a new figure (Fig. 1) and corresponding clarifications in Section 4 of our revised manuscript to illustrate the SNN adversarial examples we consider, as well as their impact in the pixel and spike domain. In brief, since our focus is on state-of-the-art low-latency SNNs (shorter simulation length $T$), our models use direct input coding where the input pixel intensities are applied as direct current to the first convolutional layer with LIF neuron activations, and a spiking representation of the input is generated through their activations. In this SNN setting, adversarial attacks operate similar as in the ANN setting, where adversarial perturbations are obtained by computing input gradients through spike-based backpropagation through time, and then applied in the input pixel domain. These adversarial input examples then clearly result in different (adversarial) spiking activation patterns after the first convolutional layer of the network, as illustrated in Fig. 1 of the revised manuscript.
>
> >In Section 3.2, I do not follow why the authors are using the RFGSM loss if they described another one previously. In general, it is a bit hard to distinguish what is novel in the paper from what has been lifted from previous literature.
>
> We rephrased our explanations and clarified these under Section 3.2. Our robust finetuning objective follows Eq. (9), which is essentially the TRADES objective with a novel additional consideration of LIF neuron firing thresholds to be adapted in this robust finetuning stage. In order to minimize this objective with gradient descent, the regularizer term requires an adversarial example $\tilde{\mathbf{x}}$ through an inner maximization step. This inner maximization step of each iteration uses the single-step RFGSM method to obtain $\tilde{\mathbf{x}}$, as in Eq. (10). For the RFGSM method in Eq. (10), differently from its original formulation with a cross-entropy loss, we instead propose to use single-step gradient ascent for $D_{\text{KL}}(f_{\theta}(\mathbf{x}')||f_{\theta}(\mathbf{x}))$. This choice was a result of an empirical observation where we achieved better results (see Table B5 ablations for alternatives).

---

> > ### Author Response · Authors · 2024-03-05
> > **Response to Reviewer xfzq (part 2)**
> >
> > >As an example, I would ask a clarification on the novelty of Section ``Conversion of Batch Normalization". If I understand correctly, the authors are advocating (i) discarding the previous statistics, (ii) re-computing them (across time and batches) for the SNN, and (iii) fine-tune the trainable weights?
> >
> > It is correct. We rephrased our explanations and clarified this under Section 3.1, ``Conversion of Batch Normalization". Traditional conversion methods either exclude batch-norm layers from the source ANN and have high simulation lengths $T$, or absorb its parameters into the preceding layer. Our proposed method can maintain low-latency in the converted SNN by exploiting robustly pre-trained ANN batch-norm parameters within tdBN layers during the conversion. Since the ANN batch statistics are not valid in the context of the initialized SNN, we discard them.  Then tdBN re-computes these during the robust finetuning stage, where we also adjust the batch-norm affine transformation parameters accordingly. Such a robust ANN-to-SNN conversion approach that preserves batch-norm operations has not been proposed previously.

---

### Review · Reviewer_eKJA · 2024-02-19

**Summary Of Contributions:**

This study proposes a way to convert the pre-trained ANN with continuous neurons into spiking neural networks(SNNs) to enhance the adversarial robustness of SNNs. The innovation of this study is converting the batch-norm layer parameters in ANN into the corresponding batch-norm parameters in SNNs. In contrast, conventional methods omit the conversion of batch-norm layer from ANN to SNN.

**Audience:**

Yes

**Broader Impact Concerns:**

No.

**Claims And Evidence:**

Yes

**Requested Changes:**

There are some mistakes/typos in the manuscript.

- Eq. 7: the $\mathbf{v}^L(t-1)$ on the RHS misses a factor such as $\tau$.
- Text right above Eq.8: the abbreviation "tdBN" is undefined and I need to guess.

**Strengths And Weaknesses:**

Overall, the paper is well-written and organized well. The numerical experiments to show the performance of the proposed conversion methods.

### Major
It seems that the SNN is over-parameterized and has redundancy in the conversion. I see the authors adjust $\phi^l$ and $\omega^l$ in the batch norm layer, and also adjust the firing threshold in the SNN. Here the adjustable firing threshold seems redundant with $\phi^l$ and $\omega^l$. Can the SNN achieve the same performance if we didn't tweak the firing threshold?

---

> ### Author Response · Authors · 2024-03-05
> **Response to Reviewer eKJA**
>
> >It seems that the SNN is over-parameterized and has redundancy in the conversion. I see the authors adjust $\phi^l$ and $\omega^l$ in the batch norm layer, and also adjust the firing threshold in the SNN. Here the adjustable firing threshold seems redundant with $\phi^l$ and $\omega^l$. Can the SNN achieve the same performance if we didn't tweak the firing threshold?
>
> We thank the reviewer for this important comment. Although the reviewer is right in this assumption for the firing threshold parameters of LIF neurons that would come after a batch-norm operation, in our architectures there are also layers that do not utilize batch-norm before every LIF neuron activation (e.g., linear classifier layers of VGG models). In these cases the optimization problem is not identical, and adversarial firing threshold tuning is still beneficial.
>
> This comparison was done in the initial submission, in Suppl. B.3 ablation studies. We now directly refer to these experiments in Section 6.1 of the revised manuscript. Our results in Table B5 of the revised manuscript explicitly tests our conversion approach in an identical setting without trainable firing thresholds during the robust finetuning stage (see the last two rows of Table B5).
> In brief, adjusting firing thresholds together with the rest of the model weights (including batch-norm parameters) yielded models with both higher clean and robust accuracies: Ours with trainable $V_{th}^l$: 92.14/13.18 vs Ours without trainable $V_{th}^l$: 91.09/11.51 under $\epsilon_3$-PGD$^{20}$ attacks.
>
> >Eq. 7: the $\mathbf{v}^L(t-1)$ on the RHS misses a factor such as $\tau$.
>
> Eq. (7) denotes our output layer neuron behavior (layer $L$), which only accumulates weighted incoming inputs as a _non-leaky_ integrator, without generating a spike. Therefore, there is no $\tau$ factor for that layer. We clarified this in text. All preceding layer LIF neurons, however, followed the discrete time dynamics in Eqs. (1), (2) and (3), where $\tau$ is present.
>
> >Text right above Eq.8: the abbreviation "tdBN" is undefined and I need to guess.
>
> We defined tdBN as the _threshold-dependent batch-normalization_ operation [Zheng et al, 2021] in Section 2.1. However, we now also clarified this abbreviation definition again in Section 3.1 (above Eq. (8)).

---

> > ### Comment · Reviewer_eKJA · 2024-04-08
> >
> > Thanks for the reply that addresses my concerns.

---

### Review · Reviewer_gqSN · 2024-02-29

**Summary Of Contributions:**

The paper presents a study on adversarial robustness in spiking neural networks (SNNs). The authors propose an approach to convert adversarially trained artificial neural networks (ANNs) into SNNs and then perform robust finetuning on the SNNs. The conversion algorithm from ANNs to SNNs involves different steps of configuring SNNs, including initializing weights, converting batch normalization, and initializing trainable firing thresholds. They also introduce an ensemble evaluation strategy for assessing the robustness of SNNs against adversarial attacks. The experiments on different datasets demonstrate the effectiveness of the proposed method.

**Audience:**

Yes

**Broader Impact Concerns:**

None.

**Claims And Evidence:**

Yes

**Requested Changes:**

1. Please discuss the relevant attacks and compare with them if possible.
2. Please clarify the setting in Table 1 and discuss the results after adversarial training.
3. Please discuss the relevant works related to batch normalization in adversarial robustness.

**Strengths And Weaknesses:**

Strengths:
1. The conversion algorithm and robust finetuning methodology are well-designed and clearly explained.
2. The ensemble evaluation strategy enhances the evaluation process, leading to more reliable robustness assessments of SNNs, which could benefit the community.
3. The experiments are sufficient, which include the evaluation of the proposed method on various datasets and networks.

Weaknesses:
1. The authors introduce an ensemble evaluation strategy to tackle the issue of obfuscated gradients. However, some relevant effective attacks have not been discussed or compared, such as [a]. In addition, how about utilizing traditional EOT? Will EOT address the obfuscated gradients in SNNs?
2. It is weird to perform AA attack to baseline ANNs but FGSM/PGD to SNNs in Table 1 since AA is regarded as a stronger attack than FGSM/PGD. In addition, how about the results of baseline ANNs after adversarial training compared with robust finetuning SNNs?
3. Some relevant works that tackle adversarial robustness via normalization layers, such as [b,c,d], should be discussed.

[a]. Rate Gradient Approximation Attack Threats Deep Spiking Neural Networks. CVPR 2023.

[b]. Batch Normalization Increases Adversarial Vulnerability and Decreases Adversarial Transferability: A Non-Robust Feature Perspective. ICCV 2021.

[c]. Random normalization aggregation for adversarial defense. NeurIPS 2022.

[d]. Removing batch normalization boosts adversarial training. NeurIPS 2022.

---

> ### Author Response · Authors · 2024-03-05
> **Response to Reviewer gqSN**
>
> >The authors introduce an ensemble evaluation strategy to tackle the issue of obfuscated gradients. However, some relevant effective attacks have not been discussed or compared, such as [a]. [...] Please discuss the relevant attacks and compare with them if possible.
>
> We included a discussion of this work in our revised manuscript. For the revision, we performed new experiments using the proposed RGA (rate gradient approximation) attack in this study [a], based on their publicly available official implementations. In particular, we compare the gradient approximate technique of this attack with our ensemble SNN evaluation approach based on a PGD$^{20}$ scenario with robust accuracies evaluated at $\epsilon_3=8/255$. Both in our CIFAR-10 and CIFAR-100 simulations (see below), results indicate that our proposed ensemble approach yields significantly stronger robust accuracy estimates, and presents an already stronger evaluation baseline than RGA. We included these results in Suppl. B.2 as Table B4 in the revised manuscript.
>
> |      CIFAR-10 / VGG11      | Clean Acc. | PGD$^{20}_{\text{RGA}}$ | PGD$^{20}_{\text{ensemble}}$ |
> |:--------------------------:|:----------:|:-----------------------:|:----------------------------:|
> |           SNN-RAT          |   90.74    |          17.98          |           **10.06**          |
> |       Ours (Natural)       |    93.53   |           1.50          |           **0.74**           |
> |   Ours (AT-$\epsilon_1$)   |    92.14   |          16.44          |           **13.18**          |
> | Ours (TRADES-$\epsilon_1$) |    91.39   |          26.51          |           **19.65**          |
> |  Ours (MART-$\epsilon_1$)  |    91.47   |          24.53          |           **17.79**          |
>
> |    CIFAR-100 / WRN-16-4    | Clean Acc. | PGD$^{20}_{\text{RGA}}$ | PGD$^{20}_{\text{ensemble}}$ |
> |:--------------------------:|:----------:|:-----------------------:|:----------------------------:|
> |           SNN-RAT          |    69.32   |           7.72          |           **6.04**           |
> |       Ours (Natural)       |    70.87   |           5.50          |           **1.40**           |
> |   Ours (AT-$\epsilon_1$)   |    70.02   |          16.81          |           **7.64**           |
> | Ours (TRADES-$\epsilon_1$) |    67.84   |          16.16          |           **7.79**           |
> |  Ours (MART-$\epsilon_1$)  |    67.10   |          17.43          |           **8.90**           |
>
> >In addition, how about utilizing traditional EOT? Will EOT address the obfuscated gradients in SNNs?
>
> Expectation-over-Transformation (EOT) is generally applied to models that have randomized components during the forward process (e.g., Poisson input coding SNNs), in order to obtain stable input gradient estimates for the adversary. Since we experiment with SNNs that already use direct input coding (as in SNN-RAT [Ding et al. 2022]) and do not have internal stochastic components, there is no randomness involved during the forward pass of the evaluated models. Therefore, repeated deterministic gradient computations of EOT do not yield different results, hence it does not tackle the problem we are addressing with ensemble SNN attacks. On the other hand, our ensemble attacks manipulate the surrogate gradient used during each gradient computation backward pass, and it is shown to have a clearly stronger effect on evaluating SNN robustness.
>
> We should however note that we have already implemented EOT-based gradients for the Poisson input coding BNTT model evaluations due to the randomness in their forward pass (see Figure 3(b) TinyImageNet comparisons), as we have clarified in Suppl. A.4.
>
> >It is weird to perform AA attack to baseline ANNs but FGSM/PGD to SNNs in Table 1 since AA is regarded as a stronger attack than FGSM/PGD. [...] Please clarify the setting in Table 1.
>
> We would like to clarify the confusion. We did not only evaluate the ANNs with AutoAttack, but also ran several simpler ANN attacks which were provided in Table A1 of the Suppl. We now highlighted this fact in our manuscript, and also added PGD$^{20}$ evaluations of ANNs to Table A1.
>
> We should note that in Table 1 we do not intend to compare ANN versus SNN performance, since their white-box attacks can not be truly identical due to the use of surrogate gradients. Table 1 only aims to illustrate the lower bound of performance we observed under the strongest attack configurations we had (AutoAttack for ANNs, ensemble PGD attacks for SNNs), and the fact that our ANN-to-SNN conversion algorithm proportionally transfers baseline ANN robustness to the converted SNN thanks to the robust weight initialization.

---

> > ### Author Response · Authors · 2024-03-05
> > **Response to Reviewer gqSN  (part 2)**
> >
> > >In addition, how about the results of baseline ANNs after adversarial training compared with robust finetuning SNNs? [...] Discuss the results after adversarial training.
> >
> > This comparison was done in the initial submission in Table B3 of Suppl. B.2 (see below), where we evaluate ANNs and SNNs under _analogous_ PGD$^{20}$ attacks. We however missed to refer to these results in the main text. We now pointed out to these results also from Sec. 6.4 of the revised manuscript.
> >
> > It is important to note that ANN and SNN models are _not_ directly comparable under the same naive PGD$^{20}$ attack configuration, and we highlight the necessity of an ensemble approach to evaluate the SNN such that a more fair comparison can be achieved. Here, the SNN evaluations in the middle columns indicate naive BPTT-based attacks, where the adversary only uses the same surrogate gradient utilized during training. These PGD evaluations are essentially identical to those for the ANNs, but might be misleading and perceived as the SNN being more robust than the ANN. Since the surrogate gradient impacts SNN attack reliability, we are considering the ensemble PGD attack on the rightmost column. This stronger attack is necessarily not exactly equivalent to the ANN attack, but we believe this is a better basis that yields a state-of-the-art comparison of robust ANNs and SNNs.
> >
> > | CIFAR-10 / VGG11 | ANN (clean / PGD$^{20}$) | SNN (clean / PGD$^{20}_{\text{vanilla}}$) | SNN (clean / PGD$^{20}_{\text{ensemble}}$) |
> > |---------------------|-------------------------|-------------------------------------------|--------------------------------------------|
> > |   AT-$\epsilon_1$   |      93.55 / 21.15      |               92.14 / 37.42               |                92.14 / 13.18               |
> > | TRADES-$\epsilon_1$ |      92.10 / 32.94      |               91.39 / 62.03               |                91.39 / 19.65               |
> > |  MART-$\epsilon_1$  |      92.68 / 27.54      |               91.47 / 49.90               |                91.47 / 17.79               |
> >
> > | CIFAR-100 / WRN-16-4 | ANN (clean / PGD$^{20}$) | SNN (clean / PGD$^{20}_{\text{vanilla}}$) | SNN (clean / PGD$^{20}_{\text{ensemble}}$) |
> > |-------------------------|-------------------------|-------------------------------------------|--------------------------------------------|
> > |     AT-$\epsilon_1$     |       74.90 / 9.46      |               70.02 / 12.56               |                70.02 / 7.64                |
> > |   TRADES-$\epsilon_1$   |      73.87 / 10.74      |               67.84 / 12.97               |                67.84 / 7.79                |
> > |    MART-$\epsilon_1$    |      72.61 / 14.72      |               67.10 / 14.27               |                67.10 / 8.90                |
> >
> > Adversarially trained ANNs and our converted robust SNNs _can_ be reliably compared under a query-based black-box attack setting via Square Attack. These comparisons are present in Fig. 5 of the revised manuscript, which revealed that SNNs ended up being more robust than ANNs at larger number of queries accessible to the black-box adversary (CIFAR-10 at 5000 queries: Ours: 50.2\% vs. SNN-RAT: 46.3\%, CIFAR-100: 18.8\% vs. 17.5\%).

---

> > > ### Author Response · Authors · 2024-03-05
> > > **Response to Reviewer gqSN (part 3)**
> > >
> > > >Some relevant works that tackle adversarial robustness via normalization layers, such as [b,c,d], should be discussed. [..] Please discuss the relevant works related to batch normalization in adversarial robustness.
> > >
> > > We added a new paragraph to our Discussion section that addresses these works as below:
> > >
> > > ``One other innovation in this work is our method of utilizing adversarially pre-trained ANN batch-norm parameters within spiking tdBN layers, without the need to omit these from the models as conventionally done.
> > > Although batch-norm layers were found to help facilitating stable adversarial training of the deep ANNs that we consider for conversion, there has also been recent studies on ANNs that particularly explore the necessity of batch normalization layers in adversarial training.
> > > In particular, Benz et al. (2021) highlight the increasing adversarial vulnerability of ANNs that utilize batch normalization layers due to the models' emerging dependence on non-robust features.
> > > Similarly, Wang et al. (2022) proposes to improve robustness by removing batch-norm layers altogether during adversarial training, which is in principle restricted to deep residual network architectures that use stable weight initialization methods for well-behaved gradients during adversarial training.
> > > There are also more recent studies that propose to replace batch-norm layers in ANNs with more robust alternatives (e.g., aggregating several normalization layers and treating the model as an ensemble of different models (Dong et al., 2022)).
> > > Although there has been some findings regarding its positive contribution to robustness (Kim \& Panda, 2021b), the role of batch normalization in SNN adversarial robustness is currently not fully explored and to be studied in future work.
> > > We use baseline ANNs that contained batch-norm layers in our simulations, since these architectures have been the commonly used ones (e.g., VGG architectures with batch-norm), especially in state-of-the-art ANN-to-SNN conversion studies (Rathi \& Roy, 2021).
> > > In principle, our robust conversion algorithm still remains compatible with any of such developments even if the baseline ANN does not contain any batch-norm layers.''

---

### Decision · Action_Editor_9Tzh · 2024-04-10

**Recommendation:** Accept with minor revision

**Comment:**

The reviewers agreed that the paper was generally clearly written and technically sound. They had some questions and points of clarification, which they authors were able to address. As well, this paper is of interest to those researchers who study SNNs. Therefore, an 'accept' decision was reached. The authors now have to upload the final, camera ready version that incorporates the changes included for the reviewers.

**Audience:**

Researchers who study SNNs, ANN-to-SNN conversions, and adversarial robustness in SNNs will find this paper interesting.

**Claims And Evidence:**

This paper explores techniques for developing spiking neural networks (SNNs) that are robust to adversarial attack. The core technique presented here is a conversion algorithm for converting ANNs to SNNs such that adversarially trained ANN weights help the SNN to be robust, coupled with a robustness fine-tuning technique for the SNNs. The authors also develop techniques for evaluating robustness of SNNs specifically that consider spike-based computation. With these evaluation techniques the authors claim that they can create SNNs that are more robust to adversarial attacks than other approaches for creating SNNs.

The reviewers agreed that the claims made in the paper are backed up by the evidence provided, and the AE concurs.

---

> ### Author Response · Authors · 2024-04-12
> **Camera-ready version has been uploaded**
>
> We thank the Action Editor and all reviewers for their time and constructive feedback that improved the clarity of our manuscript. We have now uploaded the camera-ready version of the paper.